# The impact of rare germline variants on human somatic mutation processes

Mischan Vali-Pour [1,2], Solip Park[3], Jose Espinosa-Carrasco[4], Daniel Ortiz-Martínez[4], Ben Lehner [1,2,5✉] & Fran Supek [4,5✉]

Somatic mutations are an inevitable component of ageing and the most important cause of cancer. The rates and types of somatic mutation vary across individuals, but relatively few inherited influences on mutation processes are known. We perform a gene-based rare variant association study with diverse mutational processes, using human cancer genomes from over 11,000 individuals of European ancestry. By combining burden and variance tests, we identify 207 associations involving 15 somatic mutational phenotypes and 42 genes that replicated in an independent data set at a false discovery rate of 1%. We associate rare inherited dele-terious variants in genes such as *MSH3*, *EXO1*, *SETD2*, and *MTOR* with two phenotypically different forms of DNA mismatch repair deficiency, and variants in genes such as *EXO1*, *PAXIP1*, *RIF1*, and *WRN* with deficiency in homologous recombination repair. In addition, we identify associations with other mutational processes, such as *APEX1* with APOBEC-signature mutagenesis. Many of the genes interact with each other and with known mutator genes within cellular sub-networks. Considered collectively, damaging variants in the identified genes are prevalent in the population. We suggest that rare germline variation in diverse genes commonly impacts mutational processes in somatic cells.

[1] Centre for Genomic Regulation (CRG), The Barcelona Institute of Science and Technology (BIST), Barcelona, Spain. [2] Universitat Pompeu Fabra (UPF), Barcelona, Spain. [3] Centro Nacional de Investigaciones Oncológicas (CNIO), Madrid, Spain. [4] Institute for Research in Biomedicine (IRB Barcelona), The Barcelona Institute of Science and Technology (BIST), Barcelona, Spain. [5] Catalan Institution for Research and Advanced Studies (ICREA), Barcelona, Spain. ✉email: ben.lehner@crg.eu; fran.supek@irbbarcelona.org

Cancer is primarily a disease of mutations, alterations in the DNA sequence, which result from replication errors and/ or exogenous or endogenous DNA damaging agents[1]. Genomic instability via an increased rate of mutagenesis is a major enabling mechanism of cancer[2], because it decreases the time needed to accrue the typically 2–10 somatic mutations in driver genes that are needed to initiate tumorigenesis[3,4]. Thus, identifying genetic determinants of the variability of somatic mutation rates is important for understanding and predicting variation in cancer risk among individuals as well as for determining the mechanisms responsible for tumorigenesis. Moreover, many of the most effective cancer therapies target vulnerabilities associated with defects in specific repair pathways or mutation processes and many widely used therapeutics are themselves highly mutagenic[5–7].

During the last decade, large-scale sequencing efforts have greatly enabled the analysis of somatic mutations in tumor genomes, both via whole-exome[8] and whole-genome sequencing, either from primary[9] or metastatic tumors[10]. These studies have identified driver genes and mutations[4,11–13] and also highlighted the abundance of 'passenger' mutations. Passenger mutations do not confer a selective advantage to the cancer cell and can be used to infer the sources of mutations in that particular individual and their tumor[1,14], either exogenous (chemicals, radiation) or endogenous (e.g. DNA replication errors, spontaneous deamination of cytosine)[15].

Diverse mutation types have been analyzed in cancer genomes[14,16] including single base substitutions (SBS)[17,18] and the trinucleotide they are embedded in[17,18], double base substitutions (DBS)[19], small insertions and deletions (indels)[19], copy number alterations (CNAs)[20] and other structural variants (SVs)[21]. The extracted mutational patterns (often referred to as mutational signatures) capture biological, technical, and, in many cases, unknown sources of variation[14,21]. In addition to the number and type of mutations, the regional distribution of mutations can also be informative about the activity of mutational processes[16]. For instance, in tumor genomes in which DNA mismatch repair (MMR) is impaired, there is reduced enrichment of mutations in late replicating regions (where presumably this pathway is normally less active or accurate)[16,22]. Besides replication timing, the distribution of mutations also associate with locations of chromatin marks (e.g. H3K36me3[23,24] and H3K9me3[25]), the direction of DNA replication (leading vs. lagging strand)[26,27], the direction of transcription (transcribed vs. untranscribed strand)[26], chromatin accessibility (e.g. DNase I hypersensitive sites)[28], CTCF/cohesin-binding sites[29,30], and the inactive X chromosome[31]. Moreover, the mitochondrial genome carries mutational patterns that differ from those in the nuclear genome[32,33].

While the catalogs of variation in somatic mutational patterns and rates between individuals are substantial[18,19,24,26,34,35], the extent to which this is determined by inherited genetic variants is less well understood. Examples of inherited variants that influence mutation processes include variants that cause familial cancer syndromes[36]. These include rare putative loss-of-function (pLoF) variants in the MMR genes *MSH2, MSH6, PMS2,* and *MLH1* that predispose to early-onset cancer of the colorectum and other organs (Lynch syndrome)[37]. Variants causing Lynch syndrome have been associated with several somatic mutational patterns[38], most prominently short indels at microsatellite loci[19], but also a relative enrichment of mutations in early replicating regions[22], a replicative DNA strand asymmetry[39], and an increased number of mutations in several SNV-based cancer signatures[18] due to the inefficient repair of base–base mismatches and smaller DNA loops. In addition, individuals with damaging variants in the genes *BRCA1, BRCA2, PALB2,* and *RAD51C* have

an increased risk of breast, ovarian, pancreatic, and prostate cancer, and have distinct somatic mutational patterns[40,41] such as SBS Signature 3 mutations[17], deletions at microhomology-flanked sites[17], a copy number signature[20] and several rearrangement-based signatures[21,42]. The products of these genes function in the repair of DNA double-strand breaks (DSBs) via homologous recombination, and impairment of this pathway necessitates repair via other, more error-prone mechanisms such as microhomology-mediated end joining, which create certain mutational patterns[43]. Further, tumor genomes with inactivation in the DNA glycosylase *MUTYH*[44] or *NTHL1*[45] display specific mutational signatures. Finally, tumor genomes from individuals born with pathogenic variants in *TP53* frequently have complex chromosomal rearrangements (so-called chromothripsis)[46].

These known examples illustrate how rare inherited variants can affect somatic mutation rates in humans[38], and have motivated recent analyses aiming to identify additional variants associated with specific somatic mutational patterns. In a whole-genome pan-cancer association study[9], a previously reported association[47,48] of a common deletion polymorphism in the coding region of *APOBEC3B*, altering APOBEC-signature mutagenesis, was replicated, and another nearby quantitative trait locus (QTL) associating with APOBEC mutation burden was seen[9]. Known associations of rare variants in *BRCA1* and *BRCA2* with somatic CNA phenotypes were recapitulated[9]. In addition, an association between rare pLoF variants in the DNA glycosylase *MBD4* with an increase of C>T mutations at CpG sites was reported[9], which was also found in several independent studies[49,50]. Furthermore, in a breast-cancer-specific study, the association of rare pLoF variants with APOBEC and deficient homologous recombination (dHR) SNV mutational signatures was investigated across ancestries, however without detecting hits significant across both ancestries[51].

These examples illustrate how genome-wide analyses can be used to discover germline determinants of human somatic mutation processes. Additionally, in model organisms, genetic screens have revealed that mutations in many different genes influence mutation processes[52,53].

Here, we perform a comprehensive rare variant association study using human genome sequencing data from three large-scale projects and identify genes associating with diverse somatic mutational processes. We use a gene-based testing approach combining a burden test and a variance test, two dimensionality reduction methods to define mutational phenotypes, and we consider multiple models of inheritance and multiple in silico variant effect prediction tools. We report 207 replicating associations involving 15 somatic mutational phenotypes and 42 genes, and an additional 149 associations involving 24 phenotypes and 44 genes at a more permissive false discovery rate. Rare inherited variants in a diverse set of genes therefore contribute to inter-individual differences in somatic mutation accumulation.

## Results

**Somatic mutation phenotypes in 15,000 human tumors**. To capture inter-individual variation in somatic mutation processes, we extracted 56 mutational features from ~15,000 tumor genomes analyzed as part of the Cancer Genome Atlas Program (TCGA)[8], the Pan-Cancer Analysis of Whole Genomes (PCAWG)[9] and the Hartwig Medical Foundation (Hartwig) study[10]. These features included different types of mutational signatures based on SBS, DBS, indels, and CNAs. Additionally we considered the distribution of SBS density across the genome with respect to transcription strand, gene expression, DNA replication (both the strand and timing), chromatin state (accessibility via DNAse hypersensitivity, presence of active chromatin mark H3K36me3),

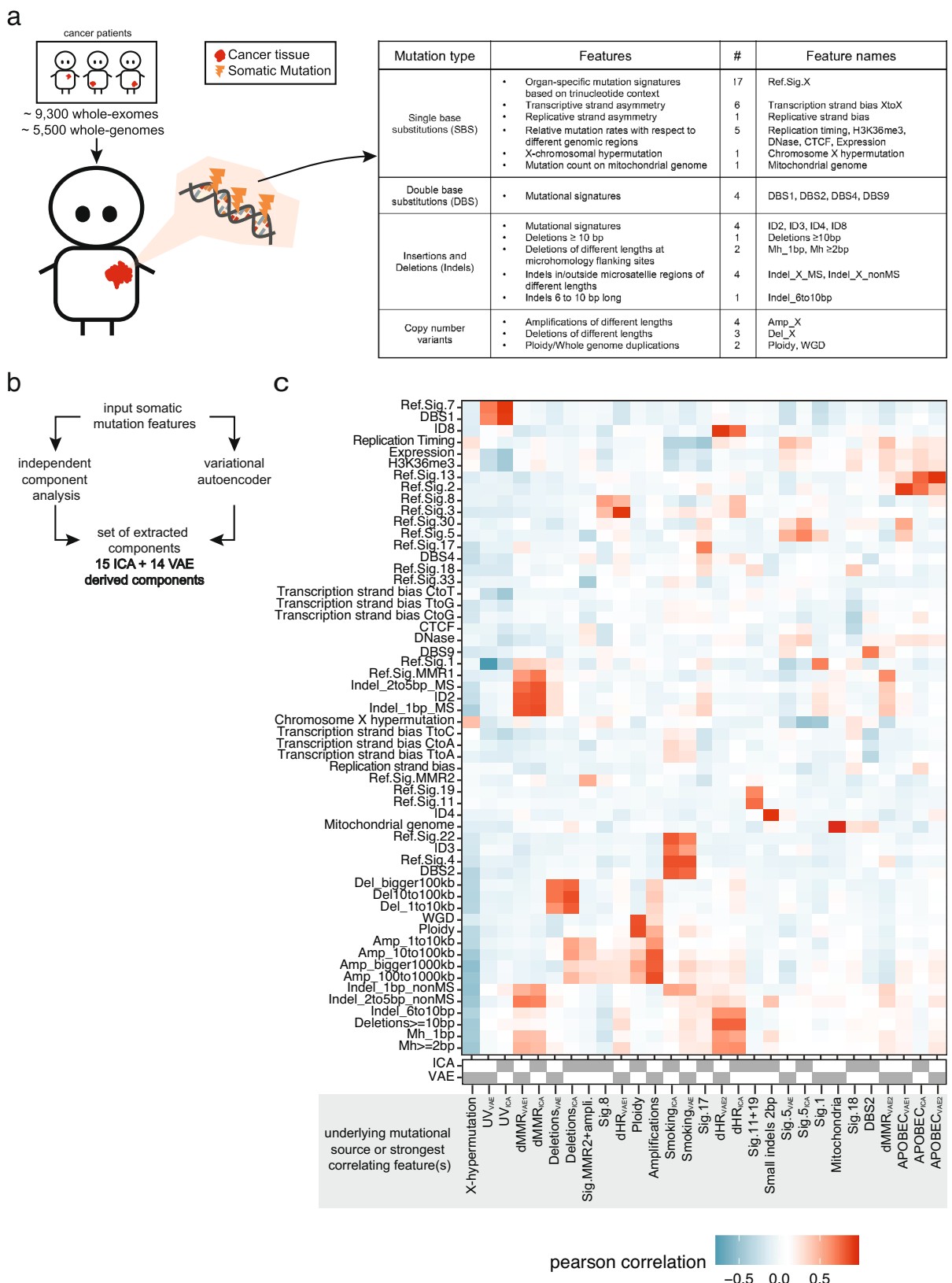

CTCF-binding sites, as well as localization on the X chromosome or in the mitochondrial genome; all of these features were previously associated with local mutation rate variability (see the "Methods" section) (Fig. 1a).

To remove the redundancy in the features, we used two different dimensionality reduction techniques—independent component analysis (ICA) and a variational autoencoder (VAE) neural network—to deconvolve the (often correlated) mutation features into mutational components (Fig. 1b). These components should both better reflect underlying causal mechanisms and their use should increase the statistical power to detect genetic associations by reducing the multiple testing

**Fig. 1 Somatic mutation phenotypes in ~15,000 human tumors. a** Somatic mutations were extracted from approximately 9300 whole-exome and 5500 whole-genome sequenced cancer genomes (left). 56 different somatic mutation features were estimated in each cancer genome, covering different types of mutations (right table). **b** Final set of somatic components was extracted by applying two methods to the input matrix (tumor samples as rows and somatic mutation features as columns): independent component analysis (ICA) and a variational autoencoder (VAE). 15 ICA-derived and 14 VAE-derived components (mutation phenotypes) were extracted. **c** Overview of extracted somatic mutation components (x-axis) and their Pearson correlation (color code) with the input somatic mutation features (y-axis). Gray strip at the bottom displays whether the component was extracted via ICA or VAE. Components were named based on the underlying mutational process or strongest correlating input feature(s). dMMR, deficient DNA mismatch repair, dHR, deficient homologous recombination. Data underlying panel **c** are provided as a Source Data file.

burden. 15 components were derived from the ICA and 14 components from the VAE (see the "Methods" section). Thirteen of the 29 components capture known mutagenic mechanisms (Fig. 1c), including UV radiation exposure ($UV_{ICA}$ and $UV_{VAE}$, including e.g. CC>TT substitutions), tobacco smoking ($Smoking_{ICA}$ and $Smoking_{VAE}$), deficiencies in MMR (dMMR; $dMMR_{ICA}$, $dMMR_{VAE1}$, and $dMMR_{VAE2}$), deficiency in the repair of DSBs via homologous recombination (dHR; $dHR_{ICA}$, $dHR_{VAE1}$, and $dHR_{VAE2}$), and APOBEC-mediated mutagenesis ($APOBEC_{ICA}$, $APOBEC_{VAE1}$, and $APOBEC_{VAE2}$). Many of the components combined different classes of mutational features. For instance, $dMMR_{VAE2}$, has a high correlation with the SNV signature RefSig MMR1[54], several types of short indels at microsatellite loci and the relative mutation rate with respect to replication timing. The remaining 16 components do not have a known mechanistic cause but can be further described via the features with which they are strongly correlated. For instance, we extracted components covering X-chromosomal hypermutation (X-hypermutation), a component covering mitochondrial SNVs (Mitochondria), and two components related to SNV-signature 5 mutations ($Sig.5_{ICA}$ and $Sig.5_{VAE}$). We note that using the outputs of both ICA and VAE methods reintroduced some redundancy in the dataset (e.g. see Fig. 1c for $dMMR_{VAE1}$ and $dMMR_{ICA}$ or for $dHR_{VAE2}$ and $dHR_{ICA}$). Since we did not know a priori which method would better capture biological mechanisms for each process, we kept the extracted components from both tools for subsequent association testing.

**Rare variant association using a combined burden and variance test.** To identify genes with rare germline variants that impact somatic mutational processes (Fig. 2a), we defined five different sets of rare pLoF variants utilizing variants with a population allele frequency of <0.1 % to extract potentially causal variants. One set involved protein-truncating variants (PTVs), two sets utilized PTVs and predicted deleterious missense variants by the tool CADD[55] at two different stringency thresholds, and the two other sets involved only those missense variants in conserved gene segments predicted by the missense tolerance ratio (MTR)[56] or by constrained coding regions (CCR)[57] method (see Methods and Fig. 2b bottom). Three models of inheritance were tested by only considering rare pLoF variants (dominant model; only germline variants), rare pLoF variants in combination with somatic loss-of-heterozygosity (LOH)[58] (additive model), and by only considering samples with biallelic inactivations of the corresponding gene (excluding genes with rare pLoF variants without somatic LOH; recessive model)[58]. In total, 15 different models were tested (Fig. 2b top). To reduce the multiple testing burden, we restricted testing to a set of 891 genes constituting known cancer predisposition genes, DNA repair and replication genes and chromatin modifiers. The combined test SKAT-O[59], which unifies burden testing and the SKAT variance test[60,61], was utilized for association testing (Fig. 2b bottom). In brief, the test statistic in SKAT-O is the weighted sum of the test statistic from a burden test and a SKAT test. Importantly, the burden test is more powerful when all rare pLoF variants in a gene are causal, while SKAT is more powerful when some rare pLoF variants are not causal or when rare pLoF variants are causal but with effects in opposite directions[59]. In SKAT-O the parameter $\rho$ indicates whether the burden or the variance test was predominantly used to identify the particular association.

**42 genes robustly associated with somatic mutation phenotypes.** Testing was performed in the discovery cohort (TCGA) across 6799 individuals of European ancestry and 12 different cancer types as well as in a pan-cancer analysis for all 15 models. Genes were only tested via the dominant or additive model when at least 2 individuals carried a rare pLoF variant in that gene. For the recessive model, genes were only tested when the gene was biallelically affected in at least two samples either by a biallelic rare pLoF variant or via a rare pLoF variant + LOH (see the "Methods" section). In total 594,462 tests were conducted. We estimated false discovery rates (FDRs) via randomization by comparing the observed p-value distribution against a random one (see the "Methods section and Supplementary Fig. 2). As an additional negative control, we considered a random set of genes, comparing the number of replicated hits at a certain empirical FDR with the random gene set to the number with our candidate gene list (Supplementary Fig. 2). It should be noted that this yields a conservative upper bound to the FDR since the random gene lists may also include genes which affect somatic mutation processes e.g. yet-undiscovered DNA repair factors.

In total, we identified 6488 associations (out of 591,302 tests) in the discovery phase at an empirical (randomization-based) FDR of 1% (Supplementary Fig. 12). Out of the 6488 hits, 3807 had a sufficient number of rare pLoF variants in the matching cancer type (see the "Methods" section) to allow re-testing in an independent validation cohort (merged PCAWG and Hartwig) in the matching cancer type, consisting of 4683 patients of European ancestry. 207 associations replicated in the validation cohort at an empirical FDR of 1%, covering 42 individual genes, 15 mutational components, 46 unique gene–cancer type pairs, and 65 unique gene–cancer type–component combinations (Fig. 3). We also checked the number of replicated associations at a more permissive FDR of 2%. At an FDR of 2%, 12,480 hits were detected in the discovery cohort, 7290 hits were able to be re-tested in the validation cohort, out of which 356 associations were replicated covering 86 individual genes, 24 mutational components, 105 unique gene-cancer type pairs, and 140 unique gene–cancer type–component combinations (Supplementary Fig. 13).

The modest validation rate (i.e. high false-negative rate) could be attributed to the low statistical power to detect medium effect-size genes, as suggested by our power analysis using the PAGEANT[62] tool (Supplementary Figs. 3 and 4). Further, by simulations that reduce the size of the validation cohort, we show that the number of replicated hits was sensitive to even small reductions in the sample size. Taken together, these analyses support that higher sample sizes will lead to the

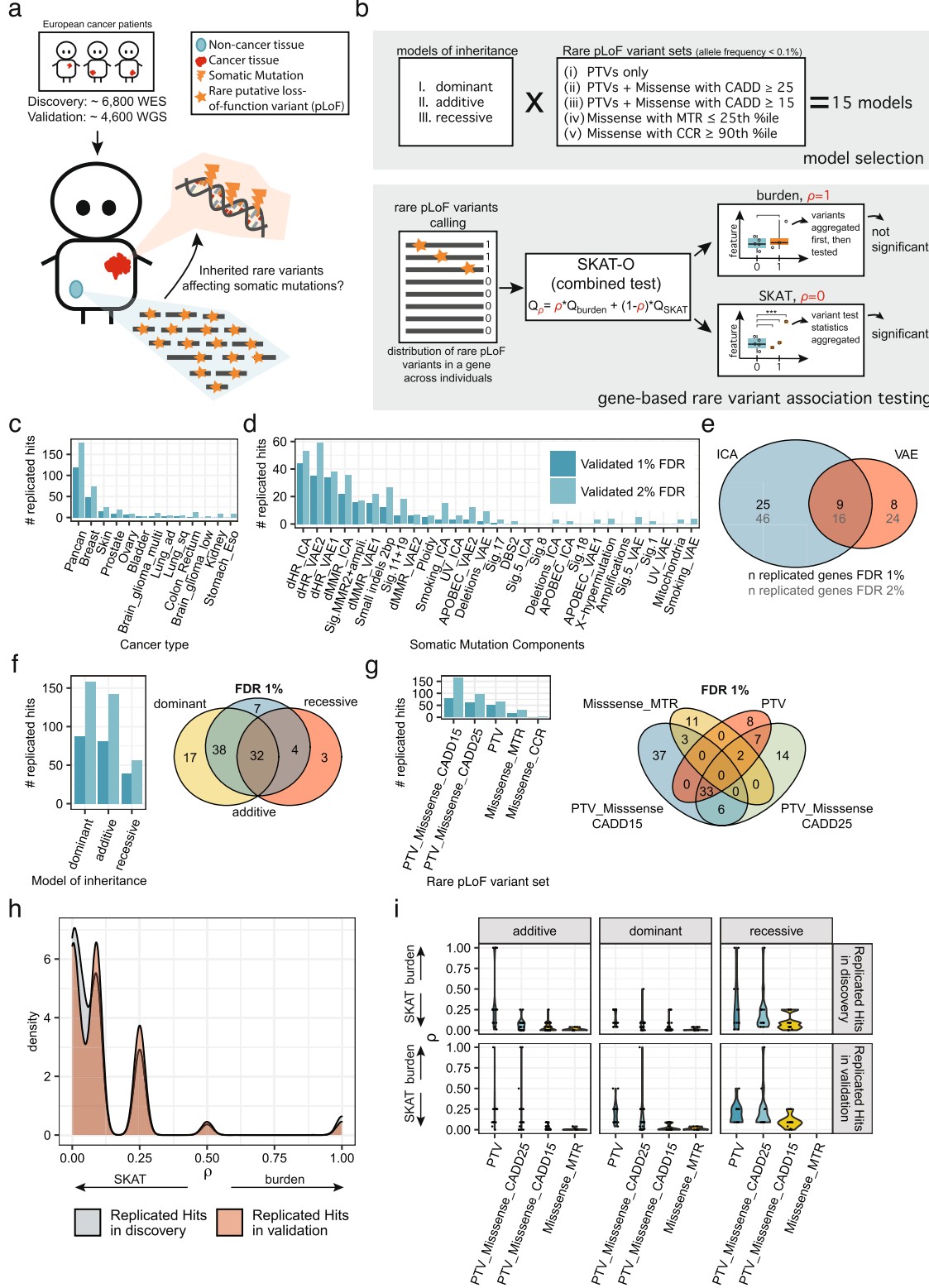

discovery and/or validation of additional associated genes (Supplementary Figs. 6–11).

Notably, seven genes associated across more than one cancer type, of which three (*BRCA1*, *EP300*, *MTOR*) associated with the same somatic mutational component across two different cancer types (Supplementary Fig. 15). Furthermore, out of the hits at a FDR of 1%, seven genes were known cancer predisposition genes[36] (*BRCA1*, *BRCA2*, *FANCC*, *MLH1*, *MSH2*, *PALB2*, and

*APC*) and at a FDR of 2% six additional cancer predisposition genes[36] were identified (*AXIN2*, *COL7A1*, *DIS3L2*, *DOCK8*, *SOS1*, and *WRN*) amongst our set, suggesting that genes that affect somatic mutation processes can also confer cancer risk.

At an FDR of 1%, most of the replicated hits were identified in the pan-cancer analysis (57%), followed by breast cancer (24%), skin cancer (7%), and prostate cancer (4%) (Fig. 2c), reflecting differential sample sizes between cancer types (Supplementary

**Fig. 2 Discovery and validation of rare putative loss-of-function (pLoF) variants associating with somatic mutation components via a gene-based combined burden and variance test. a** Associations were identified in the discovery cohort (TCGA WES) and replicated in the validation cohort (PCAWG + Hartwig WGS). **b** Associations were tested by 15 models in total, by utilizing 3 models of inheritance and 5 differently prioritized rare pLoF variant sets (all with population allele frequency <0.1%; PTVs, protein-truncating variants) (top). CADD, MTR, and CCR are different in silico variant prioritization tools. The combined test SKAT-O was applied, which calculates a weighted sum between a burden test statistic and the SKAT variance test statistic. When $\rho = 1$, the test reduces to a burden test, and when $\rho = 0$, the test reduces to the variance (SKAT) test. A schematic (not actual data) to show how the two tests can result in contrasting outcomes (bottom). **c** Number of replicated hits at a false discovery rate (FDR) of 1% and 2% across cancer types and **d** across somatic mutational components. **e** Overlap of number of genes replicating at a FDR of 1% and 2% via the two different dimensionality reduction methods. **f** Number of replicated hits at 1% and 2% FDR across models of inheritance (left) and overlap of replicated hits between models at a 1% FDR (right). **g** Number of replicated hits at 1% and 2% FDR across rare pLoF variant sets (left) and overlap of replicated hits between rare pLoF variant sets at a 1% FDR (right). **h** Distribution of $\rho$ values from the SKAT-O test (x-axis) for the 207 hits that replicated at 1% FDR, in the discovery (gray) and validation cohort (red). **i** Distribution of SKAT-O $\rho$ values (y-axis) for the 207 hits, in the discovery (top row) and validation cohort (bottom row), across models of inheritance (columns) and rare pLoF variant sets (x-axis). Data underlying panels **c–i** are provided as a Source Data file.

Fig. 12f). Furthermore, approximately half of the mutational components (15 out of 29) were associated with at least one replicated gene–cancer type pair (Fig. 2d), suggesting that many mutagenic processes are affected by germline variation in human. Many replicated hits were associated with features related with dHR (dHR$_{ICA}$: 21%, dHR$_{VAE1}$: 17%; dHR$_{VAE2}$: 16%), followed by dMMR (dMMR$_{ICA}$: 11%; dMMR$_{VAE1}$: 7%), consistent with well-established roles of HR and MMR failures in accelerating mutation rates in tumors[38]. Notably, 25 genes were only identified via an ICA derived component, while eight genes were only identified via a VAE-derived component (Fig. 2e), suggesting a complementary role of the two methods to summarize mutation processes.

Many of the replicated associations were identified via the dominant (42%) and the additive (39%) models (Fig. 2f), suggesting that heterozygous variants can alter mutation rates in humans, as was suggested for a model organism[52]. The comparatively lower number of replicated hits of the recessive model can be however largely attributed to the fact that rare pLoF variant combined with somatic LOH events are lowly frequent and thus associations could not be tested for many genes (only 4% of the 591,302 tests performed in the discovery phase were from the recessive model). Considering the proportion of replicated hits to the number of re-tested hits, the validation rate was ~2.5 times higher for the recessive model (Supplementary Fig. 12e), which was expected since many DNA repair genes are likely haplosufficient[63,64].

**Uncertainties in variant pathogenicity predictions increase the utility of a variance test over a burden test.** We further considered the number of replicated associations using different approaches and stringency thresholds for declaring a missense variant to be pathogenic. The highest number of hits replicated using the more permissive thresholds, using PTVs + missense variants at a CADD[55] score ≥ 15 (79/207, 38%), followed by PTVs + missense variants at a CADD score ≥ 25 (62/207, 30%) and PTVs only (50/207, 24%) (Fig. 2g). This suggests that some missense variants that were assigned a lower pathogenicity score —likely due to difficulties in assessing variant pathogenicity in silico[65]—can nonetheless bear on somatic mutation phenotypes. We further tested by only considering rare pLoF variants in conserved gene segments via CCR and MTR, however this yielded few replicated hits (Fig. 2g). It should be noted, however, that some hits were only identified when using the PTV-only set and were not recovered in more permissive rare pLoF variant sets.

The SKAT-O test we employed combines burden testing and a variance test component (SKAT)[59]. Examining the SKAT-O parameter $\rho$ for the 207 validated hits, in both the discovery and the validation cohort, revealed that most hits replicated via the variance test ($\rho < 0.5$ in 393/414 tests) (Fig. 2h). The variance test is the more powerful test of the two when many variants in the tested set are not causal[59]. We hypothesized that a common reason why allegedly pathogenic rare pLoF variants would not be causal is because of inaccurate prediction of damaging variants by in silico predictors[66]. If so, at the more stringent settings more hits would replicate via the burden test (which has higher power when many variants in the set are causal), while at the less stringent settings more hits would replicate via the variance test (which is robust to inclusion of non-causal variants). Indeed, several hits replicated via the burden test when using the most stringent rare pLoF variant set (PTVs only; Fig. 2i), including *MLH1*, *BRCA1*, and *BRCA2*. For the more permissive rare pLoF variant sets, the number of hits replicating via the burden test decreased and all of the replicated hits had a $\rho$ lower than 0.25 (meaning, they used nearly exclusively the variance component) for the rare pLoF variant set including missense variants at CADD ≥ 15. The positive control genes *BRCA1* and *BRCA2* still replicated in the PTV + missense CADD ≥ 15 rare pLoF variant set, but with a $\rho$ of 0 (variance test exclusively used), suggesting that this variant set included many non-causal variants.

In summary, many hits were recovered even with the more permissive rare pLoF variant sets by utilizing the combined testing approach of the SKAT-O method, suggesting the variance (SKAT) test can partially compensate for the inaccuracy of the in silico predictors. Most of the replicated hits would not have been identified by use of classical burden testing in a data set of this size.

**Genes associating with mutational patterns of defects in homologous recombination repair.** Within the set of 207 replicated associations at an FDR of 1%, 117 (57%) involved associations of *BRCA1*, *BRCA2*, and *PALB2* with various mutational components associated with dHR (Fig. 3), consistent with the known roles of these genes in the repair of DSBs. All three genes associated with features of defective HR, such as deletions at microhomology-flanked sites (dHR$_{ICA}$ and dHR$_{VAE2}$) and SNV signature 3 mutations (dHR$_{VAE1}$). In addition, *BRCA1*, but not *BRCA2*, associated with the component "Sig.MMR2 + ampli.", reflecting an increased number of amplification events. This is in accordance with a recent report, in which *BRCA1*-type dHR vs. *BRCA2*-type dHR were differentiated via the presence of duplication events[40].

We also detected additional genes associating with these dHR mutational components. In skin cancer, *PAXIP1*, *EXO1*, and *RIF1* associated with dHR$_{VAE1}$, the component correlating with SNV signature 3 mutations. In support of this, *PAXIP1* and *RIF1* have been implicated in the repair of DNA DSBs[67–69] and interact with each other[70]. Thus, these associations suggest that individuals

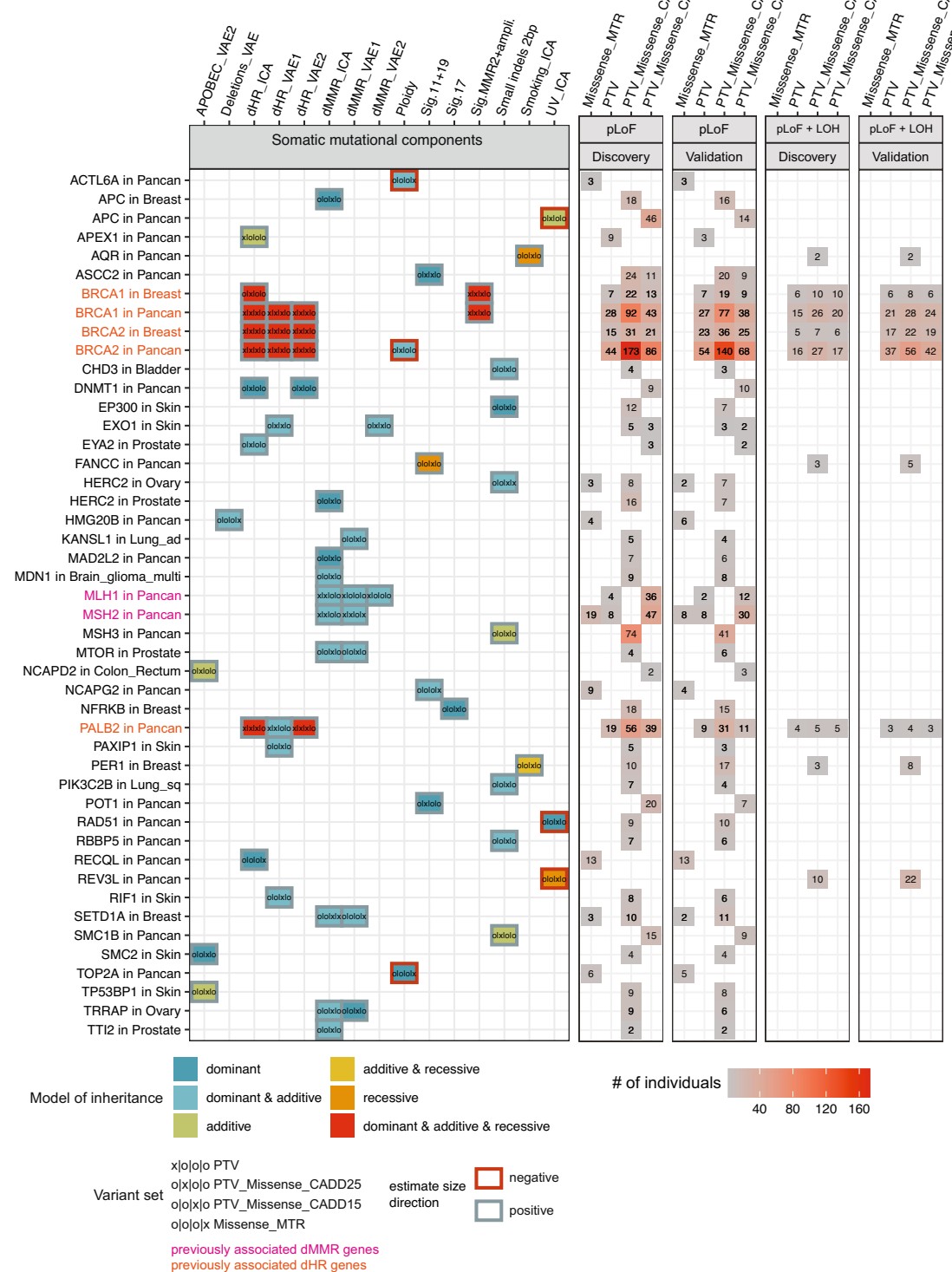

**Fig. 3 Overview of replicated associations at a false discovery rate (FDR) of 1%.** Gene-cancer type pairs (*x*-axis) and the somatic mutational component (*y*-axis 1st column) for which the association replicated at a 1% FDR. Corresponding rare putative loss-of-function (pLoF) variant set(s) are shown in each tile, where symbols and color code denote model(s) of inheritance by which they associated. Further, each tile shows the number of individuals carrying rare pLoF variants (2nd and 3rd column) or rare pLoF variants + somatic LOH (4th and 5th column) for the corresponding rare pLoF variant set associated. Gene–phenotype associations that have been previously identified are highlighted (pink for deficient DNA mismatch repair (dMMR) and orange for deficient homologous recombination (dHR)). Underlying data are provided as a Source Data file.

carrying damaging variants in either gene have an increase in signature 3 mutations, potentially reflecting a downstream effect of disrupted DSB repair. Additionally, *EXO1* knockout in a cell line model[71] was reported to result in a mutational signature correlating with signature 3 (Pearson $R = 0.71$) and signature 5 ($R = 0.71$)[71], supporting our association observed in tumors.

Furthermore, we identified pan-cancer replicated associations of *APEX1*, *RECQL*, and *DNMT1* with dHR$_{ICA}$ (with *DNMT1* additionally associating with dHR$_{VAE2}$). These associations with a microhomology deletion mutation phenotype are diagnostic of an increased activity of the microhomology-mediated end joining (MMEJ), a highly error-prone DSB repair pathway, suggesting that variants in these genes may disrupt normal functioning of the less error-prone HR and/or NHEJ pathways.

Five additional genes (*ATR*, *JADE2*, *SMARCAL1*, *TIMELESS*, and *WRN*) were identified at a more permissive threshold, associating with at least one dHR-related component (dHR$_{ICA}$ and/or dHR$_{VAE2}$). Notably, ATR and WRN proteins physically interact with BRCA1 (Fig. 4e) and play known roles in repair of DSBs[72–74], which would support these associations. In particular, pathogenic recessive variants in *WRN* cause Werner syndrome[75] and it has been suggested that the WRN helicase is crucial for the repair of dMMR-associated DSBs[76,77]. Additionally, SMARCAL1 and TIMELESS proteins directly interact with ATR (Fig. 4e).

Our analyses therefore replicate well-known associations between rare inherited variants in HR genes and dHR-like somatic mutational components, as well as identifying associations with additional genes.

**MTOR and interacting protein variants, as well as some chromatin modifiers, associate with mismatch repair phenotypes**. In the context of Lynch syndrome, germline variants in *MLH1*, *MSH2*, *MSH6*, and *PMS2*[78] affect somatic mutation patterns via an impairment of the DNA mismatch repair pathway, observed as microsatellite instability (MSI, indels at simple DNA repeats)[79,80]. MSI was also associated with SNV mutational signatures[18], as well as with a redistribution of mutations across DNA replication timing domains[22]. In accordance with this, we detected associations of rare pLoF variants in *MLH1* and *MSH2* with multiple dMMR-related components, i.e. those having a high contribution of small indels at microsatellite loci (dMMR$_{ICA}$ and dMMR$_{VAE1}$), and with the SNV-derived signature MMR1 mutations and replication timing (dMMR$_{VAE2}$; for *MLH1*).

Beyond the known Lynch syndrome genes, we also discovered associations between variation in *EXO1*, which has an established role in MMR[81] and increases the frequency of 1 bp indels when inactivated in cultured cells[82], and dMMR$_{VAE1}$ and dMMR$_{VAE2}$. However, *EXO1* also associated with dHR-related components, suggesting a more pleiotropic role for the EXO1 exonuclease in shaping somatic mutational processes in human tumors. Consistent with the association with dHR components, it was reported in yeast as well as human cell lines that *EXO1* processes DSB ends[83] and is required for the repair of DSBs via HR[84].

Multiple other genes were associated with dMMR phenotypes (all associated with dMMR$_{ICA}$ and dMMR$_{VAE1}$), including the chromatin-modifying enzyme genes *TRAAP* in ovarian and *SETD1A* in breast, and the growth signaling gene *MTOR* in prostate cancer (and in stomach + esophagus cancer with dMMR$_{VAE1}$ only at a FDR of 2%). Additionally, *TTI2* in prostate, *APC* in breast, *MAD2L2* in pan-cancer, *HERC2* in prostate, and *MDN1* in brain cancer associated with the mutation component dMMR$_{ICA}$. There is additional evidence supporting these associations for some of these genes from prior studies. *MTOR* was identified as one of four genes that regulate MSH2 protein stability[85]. Thus, a possible mechanism explaining the identified

association of *MTOR* with dMMR-linked components could be a decreased stability of MSH2 leading to dMMR and consequently, an increased number of indels. A similar mechanism could be speculated for *TTI2*, which binds *MTOR* via the TTT complex (*TELO2–TTI1–TTI2*) and is important for mTOR maturation[86]. This hypothesis is further supported by *TELO2* associating with the same component (dMMR$_{ICA}$) in kidney cancer at a more permissive FDR of 2% (Supplementary Fig. 14). Furthermore, *SETD2* associated in colorectal cancer with dMMR$_{VAE1}$ at a FDR of 2%. It has been shown in previous studies[23,24], including in cancer genomes[24], that the encoded methyltransferase SETD2 regulates MMR activity by recruiting the MSH2–MSH6 complex to H3K36me3-marked chromatin regions. Thus, this association is supported by strong evidence from prior biochemistry and genomic studies[23,24] (Supplementary Note 1.1).

Taken together, we recovered known associations of MMR genes with somatic mutational patterns and identified additional genes where germline variants are associated with MMR phenotypes, suggesting that a broad network of genes cooperates to maintain MMR efficiency in human cells.

**MSH3 and additional genes associate with a distinct dMMR phenotype enriched in ≥2 nt indels**. Interestingly, we identified associations between rare pLoF variants in several genes and a somatic mutational component ("Small indels 2 bp") that reflects indels of a size of 2 bp and longer, which is in contrast to the predominantly 1 bp long indel genomic signature caused by standard dMMR (reviewed in ref. [87]). Furthermore, this component does not have a contribution from SNV features, indicating that it is specifically capturing indels (Fig. 1c and Supplementary Fig. 27). Among others, the MMR gene *MSH3* associated with this component in the pan-cancer analysis. In contrast to the DNA mismatch repair genes *PMS2*, *MLH1*, *MSH2*, and *MSH6*, germline variants in *MSH3* have not been identified in patients with Lynch syndrome, even though they were reported to increase cancer risk[37]. The MSH2–MSH3 (MutSβ) complex has a role in repairing insertion/deletion loops rather than for base–base mismatches[88–90]. This is in contrast to the MSH2–MSH6 (MutSα) complex, which repairs base–base mismatches and indels shorter than 2 nucleotides[91,92]. These prior mechanistic studies support our association and suggest that loss of *MSH3* in cancer cells results in an increased rate of accumulation of indels of 2 bp and longer. Other genes associating with this component were *CHD3* in bladder cancer, *HERC2* in ovary cancer, *PIK3C2B* in lung squamous cell cancer, *EP300* in skin cancer (and breast cancer at a FDR of 2%), *RBBP5* in pan-cancer, and *SMC1B* in pan-cancer. Additionally, *MLH3* associated with the same component at an FDR of 2%. The MLH3 protein is a paralog of MLH1 that interacts with other MMR proteins (Fig. 4e) and was previously associated with microsatellite instability[93].

Overall, we detected associations between germline variants in *MSH3* and several other genes, and somatic indels of at least 2 bp, suggesting a causal role for MSH3 variants in a specific subtype of MMR failure which does not markedly increase SNV rates.

**Variants in APEX1 associate with increased levels of APOBEC mutagenesis**. We discovered and replicated associations between *APEX1* and three different somatic mutation components. *APEX1* encodes for a apurinic/apyrimidinic (AP) endonuclease that cleaves at abasic sites, which can be formed spontaneously or during base excision repair pathway by a DNA glycosylase[94]. At a FDR of 1%, *APEX1* associated with dHR$_{ICA}$ in pan-cancer, and at a FDR of 2% it associated with dHR$_{VAE2}$ in pan-cancer and with APOBEC$_{VAE2}$ in stomach/esophagus cancer. The somatic

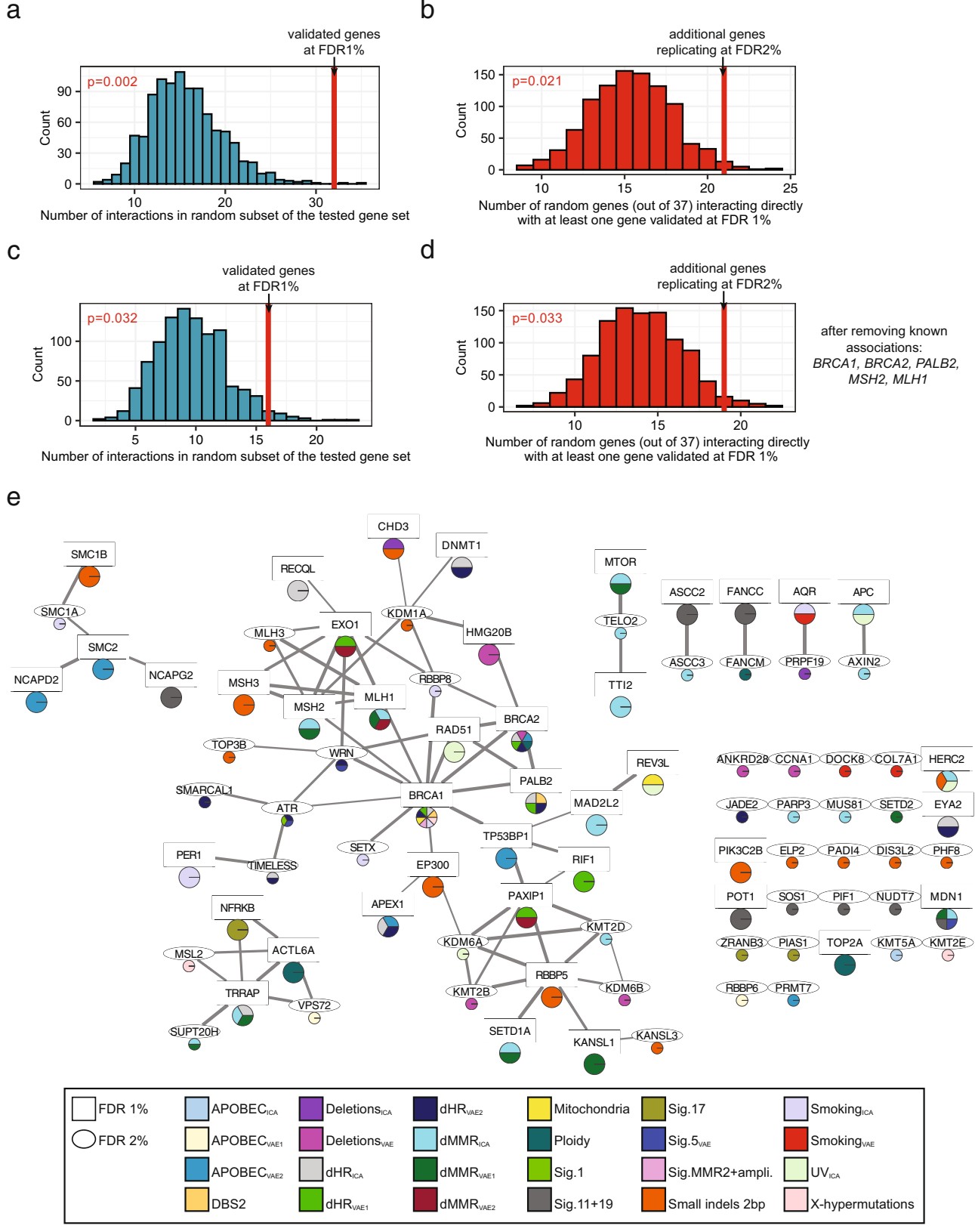

components dHR$_{ICA}$ and dHR$_{VAE2}$ are enriched for deletions at microhomology-flanked regions. Prior studies showed that the encoded protein APE1 protein plays a role in the repair of DSBs and that depletion of APE1 leads to a decrease of HR-directed repair[95], suggesting a higher reliance on alternative pathways.

The APOBEC$_{VAE2}$ component is enriched for SNV signature 13 (C>G) mutations[18]. These can be formed when the APOBEC-induced uracil is excised via the uracil–DNA glycosylase UNG and a cytosine is inserted opposite the abasic site by the mutagenic translesion DNA synthesis enzyme REV1[96].

**Fig. 4 Network analysis supports the role of rare germline variation in somatic mutational processes.** Data in all panels were generated using physical protein interactions from the STRING database that have a combined score ≥ 80%. In panels **a**–**d**, p-values were calculated via randomization using a one-sided test. **a** Number of physical interactions in a random subset of the tested gene set, controlled for interaction node degree (x-axis, blue bars). Red line shows the number of interactions within genes which replicated at a 1% false discovery rate (FDR). **b** Number of randomly selected genes from the tested gene set interacting with at least one gene, which replicated at a 1% FDR (x-axis, red bars), controlled for interaction node degree. Red line shows the number of genes, out of the ones which additionally replicated at a 2% FDR, interacting with at least one gene replicating at a 1% FDR. **c**–**d** Same as panels **a** and **b**, after excluding known genes from the analysis (*BRCA1, BRCA2, PALB2, MSH2,* and *MLH1*). **e** Visualization of physical interactions between proteins corresponding to genes replicating at 1% FDR (square) and at 2% FDR (ellipse). Color code in pie chart shows the somatic mutation components the corresponding gene was associated with (bottom panel). Line width corresponds to combined (experimental, database, and text mining) STRING interaction score. Data are provided as a Source Data file.

Conceivably, a mechanism underlying the higher burden of C>G mutations in tumors of individuals with inherited damaging variants in *APEX1* could be due to a decreased activity leading to a slower repair of the abasic site and consequently, a preference for lesion bypass via the error-prone REV1.

**Network analysis reinforces the role of rare germline variants in somatic mutation processes**. The previously known dHR genes encode proteins that physically interact as parts of the same protein complexes[97]. Similarly, the products of the known dMMR genes also physically interact[98]. We used protein–protein interactions curated in the STRING[99] database to test whether the genes in which rare germline variants associated with somatic mutational phenotypes also encoded physically interacting proteins. Such guilt-by-association network analysis has been used to support associations between somatic mutations and cancer[100,101] and between common variants and disease phenotypes[102] but has not yet been widely adopted for the analysis of rare variants.

We first considered genes associated with somatic mutation phenotypes at a FDR of 1%. These genes are strongly enriched for encoding proteins with physical interactions (Fig. 4a; median difference from a random distribution = 17 and P = 0.002 by randomization, controlling for interaction node degree). This also held true after removing dMMR/dHR genes with previously reported associations (Fig. 4c; median difference = 7 and P = 0.032 by randomization).

Secondly, we considered the 44 genes with moderate statistical support of association with somatic mutation phenotypes (those replicating at a FDR of 2%). 21 of the encoded proteins interact with at least one of the proteins encoded by the more stringent FDR 1% genes. This is again higher than expected by chance (Fig. 4b; median difference = 6 and P = 0.021 by randomization), further prioritizing these 21 genes for additional study. This also held true after removing previously known genes (Fig. 4d; median difference = 5 and P = 0.033 by randomization). Similar results were seen using the HumanNet gene network[103,] which incorporates many data sources to predict functionally-related genes (Supplementary Fig. 19).

Thus, genes with replicated associations with somatic mutation phenotypes preferentially encode proteins that physically interact in cellular networks. Genes replicating at a more permissive FDR also often connected to the same sub-networks, illustrating the potential for network-based analyses to provide supporting evidence in rare variant association studies.

To further prioritize identified genes based on their functional consistency, we made use of the networks of protein–protein interactions to test (i) the strength of interaction each protein has with known dMMR/dHR proteins (Supplementary Fig. 20) and (ii) the strength of interaction each protein has with its direct neighbors amongst the discovered set of genes (Supplementary Fig. 21). Some of the hits have high interaction scores with known dMMR/dHR genes, such as *RBBP8, MSH3, RAD51, MLH3, TP53BP1, EXO1,* and *WRN*, suggesting a higher priority for

follow-up for these hits based on functional interaction data (Supplementary Fig. 20).

**Population prevalence of damaging germline variants in genes associated with somatic mutational phenotypes**. To better estimate the contribution of rare pLoF variants to differences in somatic mutational processes, we counted how many individuals in our cancer patient datasets had certain rare pLoF variants and compared this (i) to randomly selected protein-coding genes while controlling for gene length and (ii) to the frequency of the pLoF variants in the control, largely non-cancer-patient set of gnomAD[104] (Fig. 5). Considering known mutator genes, 0.6% had PTVs in Lynch syndrome dMMR genes (*MSH2, MLH1, MSH6, PMS2*), and 1.3% had PTVs in in dHR genes (*BRCA1, BRCA2, PALB2, RAD51C*) in the discovery cohort (TCGA). Considering only the associated genes excluding known dMMR and dHR mutator genes, 1.4% had a PTV in genes that replicated at a FDR of 1%, and 2.1% in genes that replicated at a FDR of 2%. A similarly high prevalence of damaging variants in the discovered genes, relative to known mutator genes, was seen in prioritized missense variants via CADD at stringent (≥25) and permissive thresholds (≥15; Fig. 5). Additionally, when comparing this with prevalence of deleterious variants in control sets of length-matched genes, as well as the frequency of the same variants in gnomAD, there was an excess of damaging missense variants in the known dHR and dMMR genes as well as in the discovered genes at 1% and 2% FDR thresholds (excluding known dMMR and dHR mutator genes; Fig. 5), suggesting possible roles in cancer risk for these sets of genes, and also considered individually (Supplementary Fig. 18).

Taken together, these results suggest that the candidate mutator genes are affected by deleterious variants in a broadly similar fraction of the population of cancer patients like the known human germline dMMR and dHR genes.

## Discussion
We have shown here that rare inherited variants in diverse genes associate with different mutational processes. Our approach incorporated a variance-based test via SKAT-O[59], two different dimensionality reduction algorithms to extract somatic mutation patterns, the usage of different in silico variant prioritization tools[55–57], and the use of different models of inheritance for association testing. This experimental design allowed us to identify multiple replicating associations between genes and somatic mutation phenotypes. The inclusion of several genetic models and variant sets was also applied in a recent large-scale multi-disease study, which demonstrated how this approach can result in associations[105].

Most of the associations we identified were replicated only via the variance-based test SKAT, which suggests that the set of variants predicted to be damaging still contains many non-causal variants. This suggests that SKAT can help compensate for inaccuracies in current variant pathogenicity prediction tools

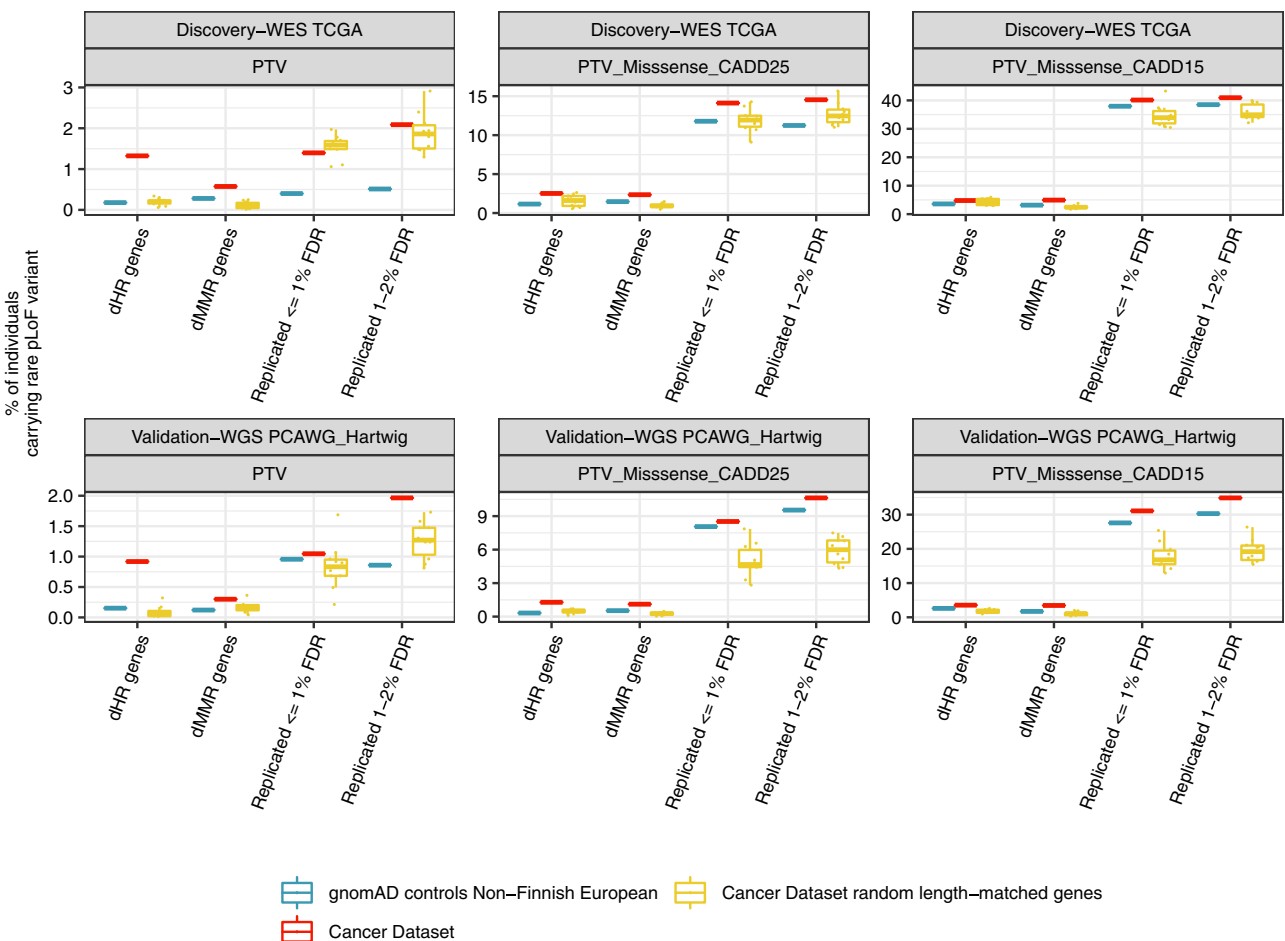

**Fig. 5 Frequency of rare putative loss-of-function (pLoF) variants across cohorts in comparison to individuals from gnomAD.** The frequencies of rare pLoF variants within the individuals (y-axis) of the discovery cohort (TCGA-WES; $n = 6799$ individuals) and the validation cohort (PCAWG + Hartwig-WGS; $n = 4683$ individuals) (rows) across different variant sets (columns) for different gene sets (x-axis). Known deficient homologous recombination (dHR) gene set includes *BRCA1*, *BRCA2*, *PALB2*, and *RAD51C*, known deficient DNA mismatch repair (dMMR) gene set includes *MSH2*, *MSH6*, *MLH1*, and *PMS2*, the replicated 1% false discovery rate (FDR) set includes genes replicating at a FDR of 1% after excluding known dMMR and dHR genes, and the replicated 2% FDR only set includes all remaining genes that replicated at a FDR of 2%. Different pLoF variant sets include protein-truncating variants (PTVs) only, and PTVs plus missense variants defined as damaging based on the in silico prediction tool CADD (thresholds at $\geq 25$ or $\geq 15$). Color code shows frequency of individuals carrying rare pLoF variants for the gene sets in the utilized cancer genomic datasets (red), for matching variants in control samples from gnomAD dataset with non-Finnish European ancestry (blue), and for length-matched randomly selected protein-coding gene sets in cancer datasets (yellow). Random selection for length-matched genes was performed 10 times, and distribution shown in boxplot. Center of each boxplot shows median, bounds of box are at 25th and 75th percentiles and minimum and maximum extend to the smallest and largest value, excluding values more than 1.5 times the interquartile range from the hinges. Only rare pLoF variants were considered that were found in gnomAD. Data are provided as a Source Data file.

and/or other technical inaccuracies such as sequencing errors. More accurate variant effect prediction tools should further increase the power of these kinds of analyses[66,106,107]. We also found that using two techniques to derive informative somatic mutation phenotypes identified more replicated associations than using either approach alone. This is consistent with findings in other fields, where different algorithms have also been found to capture complementary information, for example in gene expression analysis[108] and calling genetic variants from sequencing data[109].

We identified genes associating with dHR (e.g. *RIF1*, *PAXIP1*, *WRN*, *EXO1*, and *ATR*) and with dMMR phenotypes (e.g. *MTOR*, *TTI2*, *SETD2*, *EXO1*, *MSH3*, and *MLH3*). Several associations are supported by strong evidence from prior studies such as *EXO1* with dHR[83,84] and dMMR[71,81,82], *SETD2* with dMMR[23,24] and *MSH3* with a different form of dMMR[82,88,89]; reviewed in ref. [87]. On top of the associations with dHR- and dMMR-related

mutation components, we also identified an association of *APEX1* with APOBEC mutagenesis (as well as dHR), and additionally several genes associating with a mutational component enriched in brain and liver cancers with an unknown underlying mechanism (Supplementary Note 1.2 and Supplementary Fig. 37). Guilt-by-association network analysis has not yet been widely adopted in rare variant association studies but we found that it was useful for both connecting high-stringency replicating genes to each other and for connecting lower-confidence hits to the high-confidence genes. These interactions are useful for prioritizing the identified genes and provide specific mechanistic hypotheses connecting them to known germline mutator genes.

Interestingly, the genetic associations distinguish between two different dMMR mutational phenotypes. Firstly, the common dMMR signature, enriched for 1 bp indels and the SNV-signature MMR1; these associations involved, e.g. the Lynch syndrome genes *MSH2* and *MLH1*, and some additional genes, e.g. *MTOR*

and *SETD2*. Secondly, a distinct set of associations involved a mutational component enriched for 2 bp and longer indels, but did not encompass a notable increase in SNVs, involving the core MMR gene *MSH3*, and additionally *MLH3*, *EP300*, and *PIK3C2B*.

Furthermore, the design of our study is likely to result in a conservative bias in the number of replicated hits, because the discovery and validation cohorts were based on different sequencing technologies (WES versus WGS, respectively). WES data yields more noisy somatic mutation features, as it covers ~2% of the genome and some features (e.g. replicative DNA strand asymmetry, mutation rates at CTCF/cohesin-binding sites) are measurable at few loci and so enrichments are difficult to estimate due to low mutation counts. Moreover, the power to call germline variants at certain loci may be different for WGS and WES data. The TCGA WES data also has batch effects stemming from the different sequencing centers and sequencing technologies[110,111]. To offset this risk, we only extracted germline variants from regions with enough coverage in each of three sequencing centers, as previously[58]. This limited the number of rare pLoF variants extracted, and thus potentially also the number of discoveries.

In order to increase the sample size and thus power, we combined the cancer cohorts that contained both primary and metastatic cancers, as well as treatment-naïve and pretreated. Similarly, in the pan-cancer analyses, we aggregated data from all cancer types, with the result that the distribution of cancer types between the discovery and validation cohort was different. It is possible that some hits did not replicate due to these differences in cancer type composition.

Our initial set of somatic mutational features was largely motivated by recent reports[14,16–19,22,23,26–29,31,32,39,40,42]. Consideration of additional, complementary features could identify additional associations in future studies. Lastly, our analysis was performed on samples with European ancestry since this was the most numerous group; including sequencing data from more diverse populations is also likely to identify additional associations.

In conclusion, our findings highlight the role of rare inherited germline variants in shaping the mutation landscape in human somatic cells, leading to variability in somatic mutagenesis between individuals. The results support observations from genetic screens in model organisms suggesting that mutational processes can be affected by variation in diverse genes[52,53] and suggest that low mutation rates in human somatic cells are hard to maintain. Cooperation between many genes is required to guard against genomic instability: the canonical mutator genes (particularly MMR and HR genes) are embedded in a network of regulators and supporting genes required for optimal functioning of the DNA repair systems.

In the future, larger sample sizes with WGS data and better variant pathogenicity prediction tools will enable higher-powered association studies, further elucidating the potentially very numerous set of genes that determine human somatic mutation rates. The identification of additional genes altering human mutation processes may have important implications for understanding, preventing and treating cancer and other somatic mutation-associated disorders.

## Methods

**Study design**. In this study, the effects of rare putative loss-of-function (pLoF) variants on different somatic mutational components from cancer genomes were comprehensively analyzed. We utilized genomic sequencing data from three large-scale projects: the Cancer Genome Atlas Program (TCGA)[8], the Pan-Cancer Analysis of Whole Genomes (PCAWG)[9], and the Hartwig Medical Foundation (Hartwig)[10]. Associations between rare pLoF variants and somatic features were initially detected in the discovery cohort and hits reaching significance were re-tested in the validation cohort. TCGA WES samples were used as the discovery

cohort due to the bigger sample size and WGS samples from PCAWG and Hartwig were aggregated and utilized as the validation cohort.

### Extraction of somatic mutational features and somatic components

*Data sources in the discovery cohort*. For the somatic features which were based on SNVs, DNVs, and indels, the somatic calls from the MC3 Project[112] were used (mc3.v0.2.8.PUBLIC.maf.gz [https://gdc.cancer.gov/about-data/publications/mc3-2017]). For the somatic features based on CNVs, TCGA exome data was downloaded from the TCGA repository at NCI Genomic Data Commons [https://portal.gdc.cancer.gov/] (dbGaP accession ID phs000178) and processed as described in ref. [113]. Copy numbers were identified with the tool FACETS[114]. The tool used as input data the BAM file of the tumor sample, the BAM file of the sample-matched normal sample, and a vcf file of common human SNPs. Furthermore, 93 individuals, which were reported to be positive for human papillomaviruses in head and neck cancer samples[115], were excluded from the analysis. In total, this yielded somatic calls from 10,033 individuals.

*Data sources in the validation cohort*. Mutation calls for PCAWG were obtained from the ICGC data portal [https://dcc.icgc.org/repositories]. Somatic mutation calls and copy number calls were obtained from the DKFZ/EMBL variant call pipeline. All samples were downloaded except for ESAD-UK, MELA-AU and all project id's ending with—US in order to prevent an overlap with the discovery cohort. In total, samples from 1662 donors were downloaded. In short, single nucleotide variants were called via samtools[116] and bcftools 0.1.19[117], and indels were called via Platypus 0.7.4[118]. Copy number alterations were estimated with ACEseq v1.0.189[119] (Supplementary information in PCAWG flagship paper[9]). Data access to the estimated somatic nucleotide variants and copy number variants from the Hartwig Medical Foundation were acquired as well under request number DR-069 [https://www.hartwigmedicalfoundation.nl/en/], making up 3613 samples in total. In Hartwig nucleotide variants were called with Strelka[120] 1.0.14 and copy number alteration with the Purple tool[10]. BAM files for the melanoma dataset MELA-AU (dataset ID: EGAD00001003388; 183 individuals) and the esophagus dataset ESAD-UK (dataset ID: EGAD00001003580; 303 individuals) were downloaded from the European Genome–Phenome Archive (EGA) [https://ega-archive.org]. Somatic mutations were called via Strelka[121] 2.9.10 and copy number alterations were extracted as described above with the tool FACETS[114].

*Further processing of somatic calls*. For all datasets, regions which are known to be difficult to be aligned were excluded as well as regions which have been blacklisted by the UCSC Genome Browser[122]. As described previously[22,24] blacklisted regions by Duke and DAC were removed and the CRG75 alignability track was applied [https://genome.ucsc.edu/cgi-bin/hgTables] to only keep regions, where 75-mers in the genome can be uniquely aligned in the human reference genome hg19. Processing was performed with bedtools 2.27 [https://bedtools.readthedocs.io/en/latest/].

*Single nucleotide variants—total mutation counts*. Based on the number of SNVs in the nuclear genome, eight different somatic mutational somatic features were estimated: the total number of SNVs, the number of C>A substitutions, the number of C>G substitutions, the number of C>T substitutions in regions where the 3′ flanking site was not a G (non-CpGs), the number of C>T substitutions in regions where the 3′ flanking site was a G (CpGs), the number of T>A substitutions, the number of T>C substitutions and the number of T>G substitutions. The number of C>T substitutions was divided into two groups (at CpG sites vs. non-CpGs sites) due to the effect of CpG sites on mutation rates (due to DNA methylation)[123]. A pseudocount of 1 was added to each somatic mutational feature and all features were log transformed to the base 2.

*Single nucleotide variants in mitochondrial DNA—total mutation counts*. As other studies have pointed out, WES data can be used to extract mutations occurring in the mitochondrial DNA, due to the large amount of off-target reads[32,124]. The coverage file of each sample was used to estimate to which extent the mitochondrial genome in each sample was sequenced. Only samples in which at least 50% of the mitochondrial genome were covered by at least 4 reads were kept for further analysis. Furthermore, following a previous study[32], only variants were kept which had an allele frequency of at least 3% to remove potential false-positive calls. For the cancer cohorts Hartwig, ESAD-UK and MELA-AU, which were all based on WGS data, somatic variants in the mtDNA with a frequency of <3% were filtered out as well. After filtering, the total number of SNVs in the mtDNA in each sample was calculated. For PCAWG, mutation calls on the mitochondrial genome were downloaded from the respective study [https://ibl.mdanderson.org/tcma/mutation.html][32,33]. At last, a pseudocount of 1 was added to each individual and the feature was log transformed to the base 2.

*Single nucleotide variants—NMF-derived organ-specific signatures*. First of all, the python tool (python version 3.8) SigProfilerMatrixGenerator[125] [https://github.com/AlexandrovLab/SigProfilerMatrixGenerator] version 1.1.26 was used to generate for each dataset a matrix counting all mutations in the 96 possible trinucleotide contexts by considering the adjacent 5′ and 3′ base of the somatic variant

(16 trinucleotides for each SNV). Next, the organ-specific signatures, which were derived in the work of Degasperi et al. [54], were fit to each sample via the R package signature.tools.lib [https://github.com/Nik-Zainal-Group/signature.tools.lib]. For this step, organ-specific signature exposures were estimated by selecting for each sample the respective organ-specific signature set based on the tissue it was derived from. In cases in which no organ-specific signature set was existing due to its low sample size (e.g. mesothelioma, thymoma, penile, and vulva), the reference mutational signature set was used. In short, this aims to only fit signatures to a sample which were also identified in the according tissue. The tool uses a bootstrap-based method to only assign signatures to a sample when they reach a specific threshold ($p < 0.05$), otherwise they are set to 0. The goal of this approach is to decrease the probability of overfitting and miss-assignment of signatures[54]. In the discovery cohort the median fraction of unassigned mutations was 47% and in the validation cohort 15%, which is likely due to the low number of somatic mutations in the discovery cohort. To have a common set of signatures, all signature exposures were then converted to the reference signature set via the conversion matrix provided in ref. [54]. For further analysis we only kept 17 signatures, which had in the discovery and in the validation cohort an activity of >5% in at least one matching cancer type or in the pan-cancer analysis: Ref.Sig.1, Ref.Sig.2, Ref.Sig.3, Ref.Sig.4, Ref.Sig.5, Ref.Sig.7, Ref.Sig.8, Ref.Sig.11, Ref.Sig.13, Ref.Sig.17, Ref.Sig.18, Ref.Sig.19, Ref.Sig.22, Ref.Sig.30, Ref.Sig.33, Ref.Sig.MMR1, and Ref.Sig.MMR2. A pseudocount of 1 was added and each estimated signature count was log transformed to the base 2.

*Single nucleotide variants—transcriptive strand bias.* To estimate the transcriptive strand bias, the number of mutations occurring on the untranscribed strand and on the transcribed strand were calculated. This was performed by the python tool SigProfilerMatrixGenerator[125]. Based on the six possible base substitutions, six different somatic features were generated (C>A, C>T, C>G, T>A, T>C, T>G). For each one, the number of base substitutions occurring on the untranscribed strand were divided by the number of mutations occurring on the transcribed strand. A pseudocount of 1 was added to the numerator and denominator before division and the resulting quotient was log transformed to the base 2.

*Single nucleotide variants—replicative strand bias.* To estimate the replicative strand bias, replication timing data from lymphoblastoid cell lines was downloaded [http://mccarrolllab.org/resources/][126]. The fork polarity, which is a derivative of the replication timing estimate, was estimated as described by Seplyarskiy et al. [127]. In brief, the slope/derivative at each coordinate of the replication timing landscape was calculated by considering the region approximately ±5 kb of the coordinate. The fork polarity value reflects whether the reference strand is more likely to be replicated as the leading strand (fork polarity > 0) or as the lagging strand (fork polarity < 0). Next, the genome was divided into equal sized bins of the length of 10 kb and the average fork polarity in each bin was calculated. Further, the whole genome was split into 10 equal-sized bins based on the fork polarity estimate. To calculate the replicative strand bias, we only considered the two lowest bins (reference strand more frequently replicated as the lagging strand) and the two highest bins (reference strand more frequently replicated as the leading strand). From the perspective of the reference strand, we divided the total number of T>C, T>G, G>A, and C>A mutations occurring on the leading strand by the total number of T>C, T>G, G>A, and C>A mutations occurring on the lagging strand. This would mean for instance that a A>G mutation occurring on the leading strand was counted as a mutation occurring on the lagging strand (since T>C on the other strand). We focused on these four mutation types since replicative strand biases have been previously reported for these in connection with a deficiency in DNA mismatch repair[39]. This feature was only calculated in samples, in which at least 20 of the 4 single substitutions types were counted within the covered region. The estimated values were log transformed to the base 2.

*Single nucleotide variants—X-chromosomal hypermutation.* For generating a somatic mutational feature for X-chromosomal hypermutation[31], first of all the total number of single-nucleotide variants per megabases (MB) on each chromosome was counted. Next, the number of mutations per MB occurring on the X chromosome was divided by the average number of mutations per MB occurring on the autosomes. A pseudocount of 0.1 was added to the numerator and denominator before division and the resulting quotient was log transformed to the base 2.

*Single nucleotide variants—CTCF/cohesin-binding sites.* CTCF/cohesin-binding sites are often mutated in cancer[29,30]. To capture this somatic mutational feature, we counted the number of single nucleotide variants occurring in CTCF/cohesin-binding site and divided them by the number of mutations occurring in the flanking site (±500 bp) of the binding site. CTCF/cohesin-binding sites were obtained from Roadmap[128] and averaged over 8 cell types. Genomic regions, that were bound by CTCF in at least one cell type and by cohesin in at least two cell types were set as CTCF/cohesin-binding sites. All sites ±500 bp of the sites that were bound by CTCF in at least one cell type were set as the flanking site. Length of covered genomic regions can be found in Supplementary Table 2. This somatic feature was only estimated in samples which had at least 10 SNVs counted in total within the CTCF/cohesin binding and/or flanking site. At last, we were able to

calculate the CTCF somatic feature for 38% of the samples in the discovery cohort and 98% of the samples in the validation cohort. The ratio was log transformed to the base 2.

*Extraction of genomic region densities of expression, histone mark H3K36me3, replication timing and DNase I hypersensitive sites.* Features measuring mutation rate variation with regards to expression, histone mark H3K36me3, replication timing, and DNase I hypersensitive sites were calculated using negative binomial regression to reduce the correlation of these features with each other and to control for mutation substitution types. For this purpose, regional data from a previously published study was used[24]. In brief, levels of histone mark H3K36me3 (averaged over 8 cell types) and DNase I hypersensitive sites were downloaded from Roadmap Epigenomics[128]. Genomic regions with no signal for the corresponding feature were set as bin 0 and the remaining genomic regions were split into 5 equal-sized bins with increasing signal. In this way, genomic regions with the highest amount of histone mark H3K36me3 were put into bin 5, regions with the lowest amount into bin 1 and regions with no signal into bin 0. Replication timing information was derived from the ENCODE project using the average over 8 cell lines. Genomic regions were split into 6 equal-sized bins, where bin 1 corresponded to the latest replicating region and bin 6 to the earliest replicating region. Expression levels were based on RNA-seq data, which was obtained from Roadmap[128] and averaged over 8 cell types as well. Bin 0 represented regions with no expression (RPKM = 0) and the remaining 5 bins were split equally by increasing expression levels. All these genomic masks from ref. [24] were further processed by applying the CRG75 alignability track. For WES data specifically, the masks were intersected with the coverage mask from the MC3 project[112], since the somatic WES mutation calls were derived from there. Furthermore, the four masks (expression, histone mark H3K36me3, replication timing, and DNase I hypersensitive sites) were intersected with each other for the subsequent regression. Several bins extracted from the whole exome mask covered only a small region in the genome (<5 MB), which was expected since the exonic regions in the genome are known to be enriched for early replicating regions and histone mark H3K36me3. Since we observed that the regression often failed when bin sizes were too small, some bins were merged: replicating timing bins 1 and 2, histone mark H3K36me3 bins 1 and 2, expression bins 0 and 1, and DNaseI hypersensitive site bins 1 and 2. This step was not performed for the whole-genome masks since the covered regions for each bin were big enough. Length of covered genomic regions can be found in Supplementary Table 2.

*Single nucleotide variants—mutation enrichment calculations with regards to expression, histone mark H3K36me3, replication timing and DNase I hypersensitive sites.* The individual features corresponding to the enrichment of mutations in a particular genomic region were calculated via negative binomial regression using the function *glm.nb* from the R package MASS (version 7.3_53.1) in R 3.5.0. The regression was performed for the different features in each tumor sample as follows:

(1) mutation count ~ replication timing + mutation type + offset
(2) mutation count ~ replication timing + DNase + mutation type + offset
(3) mutation count ~ replication timing + expression + mutation type + offset
(4) mutation count ~ replication timing + H3K36me3 + mutation type + offset

In the discovery cohort (WES only) the mutation type variable had 7 possible encodings (C>A, C>T at CpG sites, C>T at non-CpG sites, C>G, T>A, T>C, and T>G), and in the validation cohort (WGS only) the mutation type variable encompassed all 96 possible substitutions within the trinucleotide context (e.g. C>A mutation within ACA context). The offset represents the nucleotide-at-risk and is the natural log of the number of nucleotides covering the respective region. As described previously[24], the coefficients obtained from the regression for the different genomic regions represent the log enrichment of mutations in each bin in comparison to a reference bin. For replication timing, the latest replicating bin was set as the reference, for expression the lowest expressing bin was set as the reference and for histone mark H3K36me3 and DNase I hypersensitive sites the bins with no signal were set as the reference. This would mean that for instance the coefficient obtained from regression (4) for bin 5 from the histone mark H3K36me3 variable describes the log enrichment of mutations in regions with a high signal of this histone mark in comparison to regions with no histone mark signal, while controlling for replication timing and the mutational context. In this way, we aimed to control for the correlation of expression levels, histone mark H3K36me3 and DNase I hypersensitive sites with replication timing and the mutational context. Especially, for WES data this approach was limited by the reduced covered genomic region and the decreased number of mutations in comparison to WGS data. The regression was only performed in samples, which had at least 30 SNVs counted. The coefficient obtained in regression (1) for the earliest replicating bin was extracted for the replication timing feature, the coefficient obtained in regression (2) for the bin with the highest amount of signal in DNase I hypersensitive sites was extracted for the DNase I hypersensitive site (DNase) feature, the coefficient obtained in regression (3) for the bin with the highest expressing regions was extracted for the expression (Expression) feature, and the coefficient obtained in regression (4) for the bin with highest amount of signal in histone mark H3K36me3 was extracted for the H3K36me3 (H3K36me3) feature.

High errors in the regression coefficients (standard error > 100) indicated that the regression failed to converge for the corresponding coefficient and thus, were removed. In the discovery cohort, 7650 replication timing coefficients, 7684 H3K36me3 coefficients, 7471 DNase coefficients and, 7664 Expression coefficients were extracted in total. In the validation cohort, 5759 RT coefficients, 5749 H3K36me3 coefficients, 5752 DNase coefficients and, 5759 Expression coefficients were extracted in total.

*Double nucleotide variants—NMF-derived signatures and fitting.* Double nucleotide variants were extracted with the python tool SigProfilerMatrixGenerator[125]. The tool counted the occurrence of 78 double nucleotide variants (AC, AT, CC, CG, CT, GC, TA, TC, TG, or TT to NN). The matrix was used as an input to extract double base substitution (DBS) signatures using the python tool SigProfilerExtractor[19] version 1.1.0 [https://github.com/AlexandrovLab/SigProfilerExtractor]. In brief, the tool uses non-negative matrix factorization (NMF) to extract mutation signatures. Since the exact number of mutation signatures is not known, the tool extracted 1–25 signatures. For each signature extraction 100 iterations were performed adding poisson noise to the samples during each iteration. For the discovery cohort the optimal solution was 3 signatures and for the validation cohort 11. Next, the tool fitted the established DBS signatures from COSMIC[13] v3.2 to the extracted de-novo signatures. Then, signature exposures were estimated by fitting the extracted COMISC signatures to each sample. In the discovery cohort the COSMIC[13] DBS signatures DBS1, DBS2, DBS4, DBS9, and DBS10 were extracted and in the validation cohort the DBS signatures DBS1, DBS2, DBS4, DBS5, DBS6, DBS7, and DBS9 were extracted. The 4 DBS signatures which were found in both cohorts were kept for association testing: DBS1, DBS2, DBS4, and DBS9. Next, a pseudocount of 1 was added to each estimated signature exposure and each estimated exposure was log transformed to the base 2.

*Insertions and deletions—total mutation counts.* Different insertion and deletion somatic mutational features were generated. First of all, the total number of indels occurring in each sample was counted. Next, the number of indels in microsatellite (MS) regions was counted due to its frequent occurrence in samples with dMMR[18,129]. For this purpose, the number of indels with a length of 1 bp and the number of indels with a length of 2–5 bp were counted within and outside MS regions. MS locations were identified via the tandem repeat search tool Phobos [https://www.ruhr-uni-bochum.de/ecoevo/cm/cm_phobos.htm]. Next, the total number of indels with a length of 6–10 bp was counted. Due to the low number of indels of this length, especially in WES data, this feature was not further split into MS vs non-MS regions. Furthermore, since deletions have often been reported to be predictive of dHR[40], different deletion features were created. The total number of deletions with a length of bigger than or equal to 10 bp was created. Also, the number of deletions at flanking microhomology sites of either 1 bp or more than 1 bp was counted by using the output matrix from the python tool SigProfilerMatrixGenerator[125]. A pseudocount of 1 was added to each feature and each feature was log transformed to the base 2.

*Insertions and deletions—NMF-derived signatures and fitting.* Small insertion and deletion (ID) signatures were extracted in the same way as described for the DBS signatures. For the discovery cohort the optimal solution was 4 signatures and for the validation cohort 10. The COSMIC[13] ID signatures were fit to the de-novo signatures and in the discovery cohort COSMIC[13] ID signatures ID2, ID3, ID4, ID7, ID8, and ID15 were extracted and in the validation cohort ID signatures ID1, ID2, ID3, ID4, ID5, ID6, ID8, ID9, ID10, ID12, ID13, and ID14 were extracted. The 4 ID signatures which were found in both cohorts were kept for further association testing: ID2, ID3, ID4, and ID8. Next, a pseudocount of 1 was added to each estimated signature exposure and each estimated exposure was log transformed to the base 2.

*Copy number variants—total mutation counts, ploidy and whole genome duplications.* Copy number-based features were generated by splitting amplification and deletion events by different sizes. The number of amplifications with a size of 1–10, 10–100, 100–1000 kb, and >1000 kb were counted. Similarly, the number of deletions with a size of 1–10, 10–100 kb, and >100 kb were counted. Next, a feature was generated based on the estimated ploidy of the tumor sample from the corresponding copy number detection tool. The number of whole genome duplication events were calculated by dividing the ploidy by 2 via integer division. A pseudocount of 1 was added to the amplification and deletion-based features, a pseudocount of 0.1 was added to the WGD feature and no pseudocount was added to the ploidy feature since ploidy can never be 0. At last, each feature was log transformed to the base 2.

*Generation of the input matrix for ICA and VAE.* For the ICA and VAE all somatic features described above were used except for the following 9 somatic features: total number of SNVs, total number of indels and total number of the 7 different single mutation substitutions types (Supplementary Figs. 22 and 23). These were excluded since they were already represented by the different NMF-derived signatures. Further, all samples were removed in which >20% of the features were not estimated due to low mutation counts. Thus, 9235/9425 samples were left in the

discovery cohort and 5597/5613 samples were left in the validation cohort. Next, missing values were replaced by the median value of the respective columns and each feature was centered and standardized to a mean of 0 and standard deviation of 1. This step was performed for the somatic features, which were extracted from three different cohorts (TCGA, Hartwig, PCAWG) separately to control for potential biases. Then, the three matrices were merged (samples as rows, features as columns).

*Independent component analysis.* The ICA was run on the 56 somatic features using the input matrix as described above. Similarly, as for the NMF, the number of ICs needs to be set before running the ICA. The methodology to extract the optimal number of components was adapted from the methodology applied previously[24] to extract the optimal number of NMF-derived components. For the extraction of ICs the R package fastICA (version 1.2.1) in R 3.5.0 was used. The ICA was run by varying the number of extracted components from 2 to 30. For each component extraction the ICA was run 200 times and the seed for the random number generator was changed before every iteration. In each iteration the ICA decomposes the input matrix into a loadings matrix (corresponding to the components and their attributed weight from each somatic feature) and a scoring matrix (also called source matrix; samples projected to component axes). After 200 iterations, the 200 loadings matrices were combined and clustered using k-medoids clustering with varying $k$ from 2 to 50. Clustering was performed with the function *pam* from the R package cluster (version 2.0.6). For each clustering the average of the mean silhouette indexes of each cluster was saved as well as the lowest and second lowest mean silhouette index of a cluster extraction. Later, extracted summary silhouette indexes for different extracted IC numbers were plotted against the different number of extracted clusters (Supplementary Fig. 24). The optimal number of components was decided visually based on the broken-stick approach (Supplementary Fig. 25). For a given extracted number of ICs, the optimal number of clusters was always times 2 since during each iteration, signs flipped randomly and thus, each component always had a correlated counterpart with opposite signs (Supplementary Fig. 26). In the end, always one component of the mirrored pair was kept. For the ICA, 15 unique ICs (using 30 clusters) were extracted. Correlations were estimated by calculating the Pearson correlation of each input somatic feature with each estimated score of each IC. Contributions were calculated by squaring the estimated loading matrix and dividing the squared loading by the sum of the loadings for the respective IC. Thus, the sum of the contributions (56 somatic input features for each IC) for each IC equals 1 (100%) (Supplementary Fig. 27).

Our rationale to apply ICA to mutational spectra to extract independent mutational components is as follows: ICA is a methodology that seeks to maximize the statistical independence of the components. Independence is considered in a general sense in the ICA methods, and is not limited to e.g. linear Pearson correlation/covariance that is minimized by PCA. Ideally, also other forms of dependency between variables would be minimized including nonlinear correlations. The implementation of ICA that we (and many others) use, the fastICA, aims to maximize the non-Gaussianity of components (as a proxy for their statistical independence). This tends to work very well for blind source separation problems, such as unmixing of sound from multiple sources, or unmixing of overlaid images, where data is expected to be non-normally distributed in each channel. Our analysis is based on the intuition that mutation process data will also similarly fit this general type of distributions, being non-Gaussian across individuals, e.g. a bimodal distribution of microhomology-flanked deletion burden resulting from individuals with active HR versus individuals with disabled HR.

*Extraction of components via a variational autoencoder.* The architecture of the VAE was adapted from studies from Way et al. [108,130]. [https://github.com/greenelab/tybalt/blob/master/tybalt_vae.ipynb], where they applied a VAE to compress gene expression data to extract biologically relevant representations. The script was modified for our purposes. In short, it is a simple ladder-VAE architecture consisting of one encoding and one decoding layer to generate a generalizable representation of the input and to use this representation to reconstruct the input. Batch normalization was performed in the encoding layer before applying the activation function *ReLu*. In the encoding layer the VAE learned a distribution of means and standard deviations to generate the latent space. This latent representation was then decoded in the decoding layer by applying the tanh function as the final activation function. Weights were initialized via the Glorot uniform initializer. We also tested adding an additional layer between the input and the encoding layer and between the latent space and the decoding layer. The extra layer always had 2 times more dimensions than the latent space and involved a batch normalization step before applying the ReLu activation function. The reconstruction loss was the sum of the mean squared error and the KL-divergence loss. To encourage learning, the ladder-VAE makes use of a so called *warm* start, meaning that it starts training without the KL divergence loss and linearly increases the contribution of the KL divergence loss after each cycle via the parameter *beta* (mean squared error + *beta*\*KL divergence loss). The linear increase of the contribution of the KL divergence loss was controlled via the parameter *kappa*.

In contrast to a previous VAE architecture[108,130], we applied the tanh function in the final decoding layer and used the mean-squared error as part of the reconstruction loss since our input was not binary. To reconstruct the input via the tanh function, all the somatic features were transformed to a range of −1 to 1 prior

to running the VAE. The data was split into 90% training data and 10% validation data and stratified by gender and cancer type. Performance was evaluated by checking the mean correlation of the reconstructed validation set with the validation input set and by calculating the correlation with selected ICs, which were shown to represent biologically relevant components. For this purpose, we calculated the maximum correlation of the components from the latent space of the VAE to the ICs $dMMR_{ICA}$, $dHR_{ICA}$, $Smoking_{ICA}$, and $UV_{ICA}$ and then calculated the average. To find the optimal hyperparameters we performed a grid search testing over 4300 hyperparameter combinations (Supplementary Fig. 29). After finding the optimal hyperparameters, the VAE was run for different latent space dimensionalities 5 times with different random initializations (Supplementary Fig. 30). In the end, the results from using a latent space with 14 dimensions was extracted for further downstream analysis using the architecture with no extra layer between input and encoder and with no extra layer between decoder and output (Supplementary Figs. 31–33).

The VAE was run in a singularity container. A docker file was generated based on the docker image *tensorflow/tensorflow:1.15.5-gpu-py3-jupyter* and the python modules *scipy*, *scikit-learn*, and *seaborn* were added. The resulting docker image was then uploaded into Docker Hub and run in a singularity container. Python version 3.6.9, keras version 2.2.4 and tensorflow version 1.15.5 were used in this environment.

We note that by applying ICA and VAE to extract components we introduced some redundancy in the dataset (e.g. see Fig. 1c for $dMMR_{VAE1}$ and $dMMR_{ICA}$ or for $dHR_{VAE2}$ and $dHR_{ICA}$). We do not think that an association with both dMMR_VAE1 and dMMR_ICA—or other such pairs of ICA and VAE components—would necessarily be considered more reliable than an association with just one member of the pair

*Estimation of tissue enrichments of components.* Tissue enrichments of individual components (Supplementary Figs. 28 and 34) were calculated as follows. For each component it was tested whether the component scores from one cancer type were significantly different to the scores of the remaining cancer types via a two-sided Welch's *t*-test. In addition, Cohen's *d* statistic was calculated between the two groups. This test was performed for each cancer type and separately for the two cohorts (TCGA and PCAWG + Hartwig). Cancer types were then grouped into their corresponding tissue of origin and the average Cohen's *d* statistic was calculated.

### Identification of rare damaging germline variants

*Extraction of rare germline variants in the discovery cohort.* TCGA bam files were downloaded from the TCGA repository at NCI Genomic Data Commons [https://portal.gdc.cancer.gov/] (dbGaP accession ID phs000178). Strelka[121] 2.9.7 was run on TCGA WES normal and tumor samples to extract germline variants. Germline variants called in the tumor samples (will be a mix of germline and somatic mutations) were used later in a downstream step to only keep germline variants which were identified in the normal and tumor tissue. In this way, we aimed to remove potential false-positive germline calls in the normal sample and to remove variants which were selected out in the tumor and thus, irrelevant for our association analysis. Germline variants which were called in the normal sample with the filter PASS were kept as well as variants which were called with the filter LowGQX but had a GQX of at least 10. Variants which were found inside gnomAD[104] v2 [https://gnomad.broadinstitute.org/downloads] with the filter PASS and had a GQX of at least 10 were kept as well as variants which were not found inside gnomAD[104], but had a GQX of at least 20. Next, variants were annotated via ANNOVAR[131] (version 2019-10-24), CADD v.1.6 scoring was added, and only exonic and splicing variants were kept. Furthermore, only variants which had allele frequency of <0.1% in gnomAD[104] (overall and in each sub-population) were kept as well as variants which were not found inside gnomAD[104]. Variants with a frequency equal to or higher than 1% within the cohort were removed. Additionally, rare germline variants were only kept when they were also found in the matching tumor sample.

*Generation of a coverage file for TCGA.* We used the same methodology as described in previous work[58] to only extract genomic regions with sufficient coverage to be sure that regions in which no damaging germline variant was called was not due to lacking coverage. In brief, within each sequencing center (BI, WU, and BCM) 100 coverage files were randomly selected. Genomic regions which were covered by at least 8 reads in 90 % of the samples within each sequencing center were kept. Next, the coverage masks of the 3 sequencing centers were intersected, making up in total a genomic mask of 60 MB in length. Only genomic regions within these sites were kept for further analysis.

*Extraction of germline variants in the validation cohort.* Germline variants from PCAWG, Hartwig, ESAD-UK, and MELA-AU were all processed in the same way if not indicated otherwise. Each cohort was processed at the beginning separately due to the different formats. The files were combined in the end. While germline calls from PCAWG and Hartwig were obtained as described above, germline variants in ESAD-UK and MELA-AU were called via Strelka[121] 2.9.10 (same approach as in TCGA), and derived from the same datasets from which the somatic calls were obtained as well. Thus, for ESAD-UK and MELA-AU the same

approach as for TCGA was applied. For PCAWG and Hartwig, germline calls with the filter PASS by the respective germline detection tool were kept. Next, variants which were found inside gnomAD[104] and had the filter PASS were kept as well as variants which were not found inside gnomAD[104] (rare singletons). Variants were annotated via ANNOVAR[131] (2019-10-24). All variants which were found inside gnomAD[104] were required to have an allele frequency of <0.1% (overall and in each subpopulation). Exonic and splicing variants were extracted. Furthermore, variants outside the CRG75 alignability mask were filtered out and variants with a frequency equal to or higher than 1% within each cohort were discarded as well. The rare germline calls from the different cohorts were combined. Further, in all cases in which germline calls were also available for the matching tumor sample, variants were filtered out if they were not found in the matching tumor sample. Germline calls for matching tumor samples were available for PCAWG, ~80% of Hartwig, and not available for ESAD-UK and MELA-AU.

*Definition of rare damaging germline variants.* In this study 5 definitions of rare putative loss-of-function (pLoF) variants were applied in addition to requiring an allele frequency of <0.1% (described above):

(5) pLoF = protein truncating variants (PTVs)
(6) pLoF = PTVs + Missense variants with a CADD55 ≥ 25
(7) pLoF = PTVs + Missense variants with a CADD55 ≥ 15
(8) pLoF = Missense variants with a missense tolerance ratio56 ≤ 25th percentile
(9) pLoF = Missense variants with a constrained coding region57 value ≥ 90th percentile

For case (5) only PTVs were considered. PTVs comprised in this study frameshift deletions, frameshift insertions, stoploss variants, stopgain variants, startloss variants and splicing variants. Splicing variants comprise the canonical splice variants annotated by ANNOVAR[131] (version 2019-10-24) and variants with a predicted donor loss or acceptor loss >0.8 by SpliceAI[132]. Pre-computed SpliceAI score files were downloaded from Illumina Basespace [https://basespace.illumina.com/s/otSPW8hnhaZR] and annotations were added to each variant (hg38 for the discovery cohort and hg19 for the validation cohort). For cases (6) and (7) potentially damaging missense SNVs were added on top of PTVs. Deleteriousness was assigned via the phred-scaled CADD[55] scores. For case (8) we only considered missense SNVs with a missense tolerance ratio (MTR)[56] [http://biosig.unimelb.edu.au/mtr-viewer/downloads] lower or equal to the 25th percentile and for case (9) we only considered missense SNVs with a constrained coding region (CCR)[57] value [http://quinlanlab.org/blog/2018/12/20/constrained-coding-regions.html] equal or bigger than the 90th percentile. On top of these variant filtering steps, two additional filtering steps were applied to all five rare pLoF variant sets in order to discard potential false-positive pLoF variants: the proportion expressed across transcripts (PEXT) metric[133] and the terminal truncating exon rule[104].

*Filtering out non-expressed variants via the PEXT metric.* The PEXT[133] score was introduced in one of the gnomAD articles and in brief, estimates to which extent a variant is expressed in a tissue based on isoform transcription levels from RNA-seq data. PEXT scores were estimated using over 11,000 tissue samples from GTEx. Thus, PEXT scores were downloaded [https://gnomad.broadinstitute.org/downloads] and added to the variant annotations. Since hg38 was used for the germline calls in the discovery cohort, PEXT annotations were first converted from hg19 to hg38 via the liftover tool from UCSC[122] (version021620). This step was not necessary for the validation cohort. Variants were only kept when they had a PEXT value >0.1 in the matching GTEx tissue. Matching a cancer type with the most appropriate GTEx tissue was mostly guided by a previous study (see Fig. 4 in Zeng et al.)[134]. For cases in which no matching GTEx tissue was available for a cancer type, the mean PEXT value was used. Exact matching of cancer types with GTEx tissues is shown in Supplementary Tables 8–10. This filter was applied to all variants not affecting splicing since many splicing variants are close to exon borders and thus, do not have a PEXT score.

*Exclusion of terminal truncating exon variants (with exceptions).* Terminal truncating variants might not have a deleterious loss-of-function effect since they can escape nonsense-mediated decay and still be functional. For these reasons, they have been also removed in the loss-of-function transcript effect estimator (LOF-TEE) of gnomAD[104]. Hence, variants occurring in the terminal exon were removed. This filter was not applied in cases in which the variant was predicted to have a deleterious effect by CADD[55] ≥ 15 or in cases in which the variant was predicted to have a splicing effect. In this way, we aimed to reduce the risk of losing potentially harmful variants, which as described in the gnomAD flagship paper[104], can be the case when the C-terminal domain of a protein exerts a crucial function. To identify variants occurring in the last exon, gene coordinates were downloaded from UCSC[122] using the NCBI RefSeq track[135]. Exon coordinates of the last exon of the longest transcript were kept. These coordinates were then intersected with the variant coordinates to detect variants occurring in terminal exons.

### Detecting and assigning putative loss of heterozygosity (LOH)

*Detecting and assigning putative LOH in the discovery cohort TCGA.* To detect LOH, we considered the copy number calls from FACETS[114]. FACETS calls were

available for 9814 samples. We extracted all LOH and DUP-LOH calls and assigned them to genes by intersecting the extracted coordinates with gene coordinates from NCBI Refseq[135] hg38. We assigned LOH to a gene in samples in which LOH was called via FACETS+ the variant allele frequency of the rare pLoF variant was not higher in the normal sample than in the tumor sample and the variant allele frequency of the rare pLoF variant was not >0.8 in the tumor and sample-matched normal sample. In this way, we aimed to only consider LOH events, when the putative rare pLoF variant of interest got enriched in the tumor via LOH since this was the tested hypothesis for the recessive and additive model. For 441 samples for which we did not have any FACETS calls, we assigned LOH to a gene in a sample when the difference in the variant allele frequency of the putative rare pLoF variant between tumor and normal sample was >0.25 and when the variant allele frequency of the rare pLoF variant was >0.8 in the tumor and sample-matched normal sample.

*Detecting and assigning putative LOH in the validation cohort.* For PCAWG (excluding ESAD-UK and MELA-AU), CNV calls from ACEseq[119] v1.0.189 were further processed. All passed calls with the assignments LOH, LOHgain, or cnLOH were extracted and genes were assigned to the LOH events as before (using NCBI Refseq[135] hg37). We excluded LOH calls when the corresponding rare pLoF variant in the respective gene had a lower allele frequency in the tumor than in the sample-matched normal sample and the allele frequency was not >0.8 in both tissues.

For ESAD-UK and MELA-AU, CNV calls were available via FACETS[114] and LOH was called as described for TCGA. In contrast to the steps performed for TCGA, germline calls from the tumor tissue were not available for ESAD-UK and MELA-AU. Thus, LOH calls were not further filtered.

For Hartwig, CNV calls were provided via the tool Purple[10]. LOH was assigned to locations in which the minor allele ploidy was <0.4. LOH calls were excluded in cases in which the allele frequency of the rare pLoF variant was lower in the tumor than in the sample-matched normal tissue and the allele frequency of the rare pLoF variant was not >0.8 in the normal and tumor tissue. This was only applicable to the samples in which germline calls from the tumor genome were available (678 samples with germline calls from tumor genomes not available).

### Gene-based rare variant association testing

*Extraction of common germline variants and sample-level quality control.* Common variants were extracted from the normal samples to apply some sample-level quality control as well as to prepare the data to perform a PCA for extracting population ancestry. The following steps were performed for the dis- covery cohort (TCGA) and the validation cohort (PCAWG and Hartwig) separately. Germline variants which were called with the filter PASS were kept. Also, in accordance with the extraction of rare germline variants, variants with the filter LowGQX but a GQX ≥ 10 were kept in the respective cohorts (TCGA, ESAD-UK and MELA-AU). Common variants were extracted by only keeping variants which were identified inside gnomAD[104] with the filter PASS and with an allele frequency >5% within the overall population. In TCGA all variants within the generated genomic mask were retained and in the other cohorts all variants within the CRG75 alignability mask were retained. Loci, in which more than 2 alleles existed, were removed. The total number of common variants inside each sample was calculated and within each cohort (TCGA, Hartwig, PCAWG) samples with an altered number of variants 1.5 standard deviations away from the mean were discarded (214 samples in TCGA, 212 samples in Hartwig, 204 samples in PCAWG) (Supplementary Figs. 38a, 39a and 39b). Next, common variants for each cohort were uploaded into PLINKv1.90b6.1 and further processed there. Missing genotypes were set as homozygous for the reference allele. Only variants with a MAF > 5% were retained and samples with a heterozygosity rate ±3 standard deviations away from the mean were removed (127 samples in TCGA, 54 samples in Hartwig, 39 samples in PCAWG) (Supplementary Figs. 38b, 39c and 39d). For the following steps, variants on the sex chromosomes, on the mitochondrial chromosome and within regions with high amount of linkage disequilibrium [https://github.com/meyer-lab-cshl/plinkQC/tree/master/inst/extdata] were removed. Also, variants extensively deviating from the Hardy–Weinberg-equilibrium with $p < 10^{-6}$ were excluded.

*Identification of duplicated or related individuals.* The dataset was pruned on the discovery cohort (TCGA) and on the merged validation cohort (PCAWG and Hartwig) separately, applying a window size of 50 bp, a step size of 5 and a $r^2$ threshold of 0.2. The identity-by-state (IBS) matrix was calculated for all pairs of individuals within each cohort. Within all pairs of individuals with identity-by-descent (IBD) > 0.185 (0.185 would be the expected value for individuals between third- and second-degree relatives) one individual was removed (542 samples in TCGA, and 479 samples in PCAWG and Hartwig) (Supplementary Figs. 38c and 39e).

*Extraction of European individuals.* To extract individuals of European ancestry the pruned dataset was used and a principal component analysis (PCA) was performed. The PCA was run on the discovery cohort and on the merged validation cohort (Supplementary Figs. 40 and 41). The first ten principal components were used for clustering using the R package tclust (version 1.4.2), which trimmed 1% of the outlying samples as described previously[58]. Individuals were grouped into $k = 10$ clusters and European groups were selected based on the reported TCGA/

PCAWG annotations. In total 7864 individuals were retained in the discovery cohort and 4691 individuals were retained in the validation cohort. The PCA was repeated on the pruned dataset for the individuals of European ancestry in the respective cohorts to extract the PCs, which were used as covariates in the association testing (Supplementary Figs. 42 and 43).

*Gene-based rare variant burden testing.* As described above 29 somatic mutational components were extracted from the discovery and validation cohort from the tumor genomes. rare pLoF variants were extracted from the sample-matched normal samples. Gene-based rare variant burden testing was only performed on samples which survived the quality control filters (as described above). We limited the analysis to individuals with European ancestry due to the bigger sample size. In addition, only samples were kept, in which at least 10 SNVs were counted. In total 6799 samples were left in the discovery cohort for testing and 4683 samples were left in the validation cohort for testing.

*Gene set.* For testing, RGDVs occurring in 891 different genes were extracted. The gene set covered DNA damage response genes[136] [https://doi.org/10.1038/nrc3891], known cancer predisposition genes[36] [https://doi.org/10.1038/nature12981], genes involved in chromatin organization [https://pathcards.genecards.org], genes involved in DNA double-strand repair [https://pathcards.genecards.org], genes which were reported to regulate MSH2 stability[85] [https://doi.org/10.1038/nm.2430], and human homologs of genes, in which heterozygous mutations were reported to cause genetic instability in *Saccharomyces cerevisiae*[52] [https://doi.org/10.1038/s41586-019-0887-y]. Effectively, out of the 891 individual genes 746 genes were tested in the most permissive rare pLoF variant set (7) in pan-cancer. The remaining genes were not tested in the discovery cohort since not enough rare pLoF variants were identified in these genes to test them.

*Association testing via SKAT-O.* Association testing was performed in each cancer type separately and with all cancer types together (pan-cancer). The effect of a gene on a somatic component was only tested when a rare pLoF variant in that gene was identified in at least two individuals (Supplementary Fig. 16). Testing was performed across 12 cancer types as shown in Supplementary Table 6. Accordingly, depending on the cancer type different numbers of genes were tested in total.

Association testing was conducted via the unified testing approach of SKAT-O[59]. While in burden testing the variants are aggregated first and then jointly regressed against a phenotype, in SKAT the individual variants in a gene are regressed against the phenotype (Supplementary Figs. 35 and 36), and then the variance of the distribution of the individual variant score statistics is tested. SKAT-O combines the tests SKAT[61] and burden via a weighted mean:

(10)     $Q_\rho = \rho * Q_B + (1-\rho)Q_S$.

Here, $Q_\rho$ is the final statistic from the weighted mean of the burden statistic $Q_B$ and SKAT statistic $Q_S$. The parameter $\rho$ influences how strongly each test is weighted. SKAT-O testing was performed via the R package SKAT[59] 2.0.1. For testing, the covariates were firstly regressed against the somatic components with the function *SKAT_Null_Model*. When applicable, age at diagnosis, sex, ancestry (first 6 PCs) and cancer type were used as covariates. Categorical variables were encoded as dummy variables with the R package fastDummies 1.6.3. Missing age information was imputed by taking the median value in the respective cohort. After initializing the null model, SKAT-O was run by using the function *SKAT* and setting the method to *SKATO*. The function ran SKAT-O with 10 different values of $\rho$ (from 0 to 1) and reported the $\rho$ value which led to the lowest *p*-value.

Three models of inheritance were tested in total and individual variants were encoded as follows:

(11)     Dominant: no rare pLoF = 0; rare pLoF = 1

(12)     Additive: no rare pLoF = 0; rare pLoF = 1; rare pLoF + somatic LOH or biallelic rare pLoF = 2

(13)     Recessive: no rare pLoF = 0; rare pLoF + somatic LOH or biallelic rare pLoF = 1: rare pLoF without somatic LOH = excluded sample.

Significance of a gene with a specific model would not necessarily imply that the gene follows that model of inheritance. Taken together, 3 models of inheritance were tested with 5 different rare pLoF variant sets, making up in total 15 models to test across 12 different cancer types and pan-cancer. In total, 15*12*29 = 5655 model scenarios could have been tested at most. Ultimately, 4693/5655 scenarios were tested in the discovery phase.

*Estimation of effect sizes via burden testing.* Since no effect sizes were reported in SKAT-O, we also performed gene-based burden testing (aggregating variants occurring in the same gene) applying the same models as above. Association testing was performed via linear regression with the *lm* function of the R base package stats in R 3.5.0 as follows:

(14)     Somatic Component ~ Gene + Covariates

The somatic components were coded as quantitative variables as described above. The gene variable was encoded as a binary categorical variable depending on the model of inheritance (additive, recessive, dominant). When applicable, we controlled for age at diagnosis, sex, cancer type and ancestry (first 6 PCs) as

covariates. In total, burden testing was performed for each scenario which was also tested via SKAT-O.

*Quantile–quantile plots for quality control.* To check for potential biases in testing, we plotted quantile–quantile plots (QQ- plots) for each somatic component tested for each scenario (model of inheritance, rare pLoF variant set) in the respective cancer type and calculated the corresponding inflation factor $\lambda$. For the QQ-plots, the expected *p*-value was calculated by ranking all tested genes and dividing the rank of a gene by the total number of genes tested. The idea behind the QQ-plots was that most genes were expected to not have an effect on a somatic component and thus, most *p*-values would be distributed randomly and fall on a linear line when ordered. The inflation factor $\lambda$ was calculated to check for inflation, which would be indicated by $\lambda > 1$. The inflation factor $\lambda$ was estimated by dividing the median of the chi-squared test statistic of the *p*-values by the expected median of the chi-squared distribution, which would be a chi-squared distribution with one degree of freedom. QQ-plots with no inflation would have an inflation factor of $\lambda \approx 1$ and deflated QQ-plots would have an inflation factor of $\lambda < 1$. Ultimately, we excluded model scenarios in which at least 100 genes were tested and the inflation factor was $\geq 1.5$ (19 out of 1909) (Supplementary Note 1.3 and (Supplementary Fig. 1).

*Estimation of false discovery rates.* We calculated false discovery rates (FDRs) via two approaches: empirical FDR and via a randomized set of genes. To estimate the empirical FDR, the somatic component matrix (somatic components as columns and sample IDs as rows) was randomly shuffled within each cancer type. Importantly, the link between individuals and somatic components was broken down, but the correlation structure between components was conserved. Then, with the randomized somatic component matrix, testing was performed in the same way as it was performed before. We calculated empirical FDR thresholds for each cancer type (or pan-cancer) separately. For instance, the *p*-value at which 1% of the associations from the randomized run would have been called as a hit (false discovery) corresponds to a FDR of 1%.

For our second approach, we repeated the whole analysis using 1000 random genes. We generated a list of genes, which were not in our pre-selected gene list of 891 genes and in which rare pLoF variants according to rare pLoF variant set III. were identified in at least two samples. In addition, we discarded all genes which were reported to have a physical interaction with any gene from our pre-selected gene list according to the reported physical interactions from STRING v11.5[99] [https://string-db.org/cgi/download?sessionId=bPz0GBvgDw3p] with a combined score of at least 50%. Out of 11,408 remaining genes, 1000 genes were randomly selected and used for testing. Next, we performed the same steps as it was performed for the pre-selected list of genes, including the calculation of empirical FDRs via randomization and the exclusion of model scenarios with high inflation factors (31 out of 1885). Based on the conservative hypothesis that there would be no real associations from the random list of genes, we calculated FDRs at different empirical FDR thresholds by dividing the number of hits, which were detected via the random list of genes by the number of genes detected at the same empirical FDR with our pre-selected list of genes. For instance, at an empirical FDR of 1% we identified 44 hits with our random list of genes and 207 hits with our pre-selected list of genes. Thus, we estimated a FDR of 44/207 ≈ 21% at our empirical FDR of 1%.

*Identification of associations in the discovery cohort and re-testing in the validation cohort.* Hits were identified in the discovery cohort when they were significant either at a FDR of 1% or 2% based on the estimation of the empirical FDR. These were then re-tested in the matching cancer type based on the tissue of origin (Supplementary Table 6). In total, for 12 individual cancer types a matching cancer type based on the tissue of origin was available in the validation cohort with a sample size of at least 50 samples: bladder cancer, brain glioma multiforme, low-grade glioma, breast cancer colorectal cancer, kidney cancer, lung adenocarcinoma, lung squamous carcinoma, ovary cancer, prostate cancer, skin cancer, stomach and esophagus cancer. Hits which were identified with all cancer types together (pan-cancer) were re-tested in the validation cohort in the same way. We called a hit as replicated when it reached the empirical FDR of either 1% or 2% and had the same estimate effect direction as in the discovery cohort. Effect size directions were extracted from the performed burden tests.

*Network analysis.* For the network analysis, we downloaded protein network data from STRING v11.5[99] involving only physical links, and from HumanNet[103] v3 the functional gene network (HumanNet-FN) [https://www.inetbio.org/humannet/download.php]. From STRING we only kept interactions which had a combined confidence score (based on experimental, database, and text mining) of at least 80%. The following steps were performed for each protein network separately.

Firstly, we extracted all interactions which involved interactions between genes from our pre-selected gene list of 891 genes. We calculated the total number of interactions our replicated genes had at an empirical FDR of 1% with each other. It was tested via randomization whether this number was higher than one would expect at random. For this purpose, we selected randomly the same number of genes and calculated the total number of interactions these genes had with each other. We controlled for the total number of interactions each gene had, since some genes (e.g. *BRCA1*) have in general a lot of physical interactions, which would

confound our results. To control for this, we counted the total number of interactions our replicated genes had, split them into 10 equal-sized bins, assigned all our pre-selected genes a bin, and then randomly selected the same number of genes from each bin. Randomization was performed 1000 times.

Next, we counted how many genes, which only replicated at an FDR of 2%, had at least one interaction with a gene which replicated at an FDR of 1%. Here, we applied the same approach. We counted the total number of interactions each gene, which only replicated at an FDR of 2%, had in total and split the number of interactions into 10 equal-sized bins. Each gene from our list of genes was assigned a bin and then we randomly selected 1000 times the same number of genes from each bin and performed the same calculation.

*Gene prioritization via network analysis.* Gene prioritization was performed via two approaches. Firstly, within each network (STRING and HumanNet) based on the genes replicating at a FDR of 2%, it was estimated how strongly each protein interacted with a known dHR (BRCA1, BRCA2, PALB2, RAD51C) or dMMR (MLH1, MSH2, MSH6, PMS2) protein. For this purpose, for each protein, the strongest interaction score with a known dHR or dMMR protein was extracted. Next this score was compared against the highest interaction scores of all other non-dHR/dMMR proteins with dHR/dMMR proteins from our list of 891 genes, which had a similar node connectivity. Thereby, a *p*-value was calculated and an effect size was estimated by subtracting the highest interaction score of the protein of interest from the median of the interaction scores of the other genes. Proteins more likely interacting directly with dHR/dMMR proteins got prioritized. For this analysis, the same HumanNet network as described above was used and the same STRING network as above, but without any pre-filtering (based on scores). Secondly, genes were prioritized by considering how strongly a protein interacted with a neighbor in the network (network with the genes replicating at a FDR of 2%) in comparison to a random network with a similar node connectivity and the same number of genes out of the gene list of 891 genes. For each protein, the interaction scores with its direct neighbors were considered the highest one was saved. Next, a random network was generated with the same number of proteins from the list of 891 genes with a similar node connectivity and only the tested protein was kept. Then, again the highest interaction of the tested protein with its neighbors was extracted (set to 0 if no interaction). This randomization was repeated 10,000 times. A *p*-value and effect sizes was then estimated by comparing the observed interaction score against the distribution of 10,000 interaction scores from the randomization. The same HumanNet network and STRING network as described above were utilized.

*Calculation of frequency of rare pLoF variants in length matched randomly selected genes.* To calculate the number of rare pLoF variants occurring in a control set of genes, we matched each replicated gene randomly with a gene covering the same length. For this purpose, we intersected the TCGA coverage file with the reported exonic coordinates provided by NCBI RefSeq[135] track hg38. We only considered protein-coding genes. The covered length of each gene was calculated in kilobases and each replicated gene was randomly matched 10 times with a gene, which covered the same length in our data. Subsequently, RDVGs based on different sets were counted in the replicated gene sets as well as in the length matched control genes. For the validation cohort PCAWG_Hartwig-WGS, the same approach was applied. Here, the coordinates from the CRG75 alignability track were intersected with the exonic coordinates provided by NCBI RefSeq[135] track hg19 to determine the length of the coding region for a gene.

*Calculation of frequency of rare pLoF variants in GnomAD.* To compare variant frequencies between cancer datasets (discovery and validation cohort) (Supplementary Fig. 17) and gnomAD, all variants which were detected in the cancer datasets and in gnomAD v2.1.1 were retained. Rare pLoF variants in the cancer datasets which were previously not detected in gnomAD (potentially ultra rare variants) were removed in order to remove potential biases due to technical reasons. Variant frequencies between cancer dataset were compared with gnomAD by summing the variant frequencies of the same rare pLoF variants which were identified in the respected cancer dataset in gnomAD using the reported variant frequencies in gnomAD control samples (all ancestries) and in gnomAD control samples of Non-Finnish European ancestry. Variant frequencies were summed up since no individual level data is available in gnomAD and we applied the assumption that a rare pLoF variant in the respective gene would only occur once per individual. For the cancer datasets the number of rare pLoF variants were divided by the size of the cohort to estimate the variant frequency in the respective cohort (Fig. 5 and Supplementary Fig. 18).

*Power analysis.* Power analysis was performed using the tool PAGEANT[62] [https://andrewhaoyu.shinyapps.io/PAGEANT/] with code on github [https://github.com/andrewhaoyu/PAGEANT]. Power analysis was performed for each cancer type (and pan-cancer) in each cohort (discovery and validation) separately by running PAGEANT with the respective sample size. Further, the respective thresholds for a FDR of 1% and 2% were utilized. PAGEANT was run with the SKAT simulation by varying the percentage of total variance explained (TVE) by the variants in a gene, since this number (related to the effect size of the gene) is not known a priori (Supplementary Figs. 3 and 5).

*Estimation of total variance explained for known high-effect size genes.* To have a reference point in the power analysis for the parameter percentage of phenotypic variance explained by the variants in a gene (total variance explained, TVE), we estimated the lower-bound of the percentage of the variance explained by the variants in a gene for different gene–phenotype–variant set combinations. We focussed on known genes with high effect sizes, namely dHR genes (*BRCA1*, *BRCA2*, *PALB2*, and *RAD51C*) and dMMR genes (*MLH1*, *MSH2*, *MSH6*, and *PMS2*) with the matching mutational phenotypes. The analysis was performed in the discovery cohort on pan-cancer to have the highest possible sample size. To estimate the total variance explained for a gene, first the respective phenotype was regressed in a multiple linear regression against age at diagnosis, gender, cancer type and ancestry (first 6 PCs). Next, the residuals from this regression were regressed against all rare pLoF variants (depending on the variant set: PTVs only or PTVs + Missense with different CADD thresholds) in a multiple linear regression. The total variance explained ($R^2$) from this regression can be interpreted as the lower-bound of the total variance explained by variants in a known (positive control) mutator gene, and utilized as a reference point in the power analysis. It would be expected that this number would increase with a higher sample size since more potentially rare pLoF variants could be detected.

*Subsampling analysis.* Subsampling was performed by randomly retaining 95%, 90%, and 80% of the data of the validation cohort (PCAWG + Hartwig). For each scenario, subsampling was performed three times, generating in total nine datasets. Association testing was performed in the same was described above. Empirical FDRs were estimated in the same way in order to determine which hits, that were previously identified in the discovery cohort with the complete dataset of the discovery cohort, replicated in the validation cohort.

**Reporting summary**. Further information on research design is available in the Nature Research Reporting Summary linked to this article.

## Data availability

In this study published datasets were reanalyzed. TCGA WES bam files of primary tumors and matched normal samples (dbGaP accession ID phs000178, restricted access that can be applied to following instructions on dbGaP) were downloaded from the TCGA repository at NCI Genomic Data Commons [https://portal.gdc.cancer.gov/]. Somatic mutation calls for TCGA were downloaded from the MC3 project (mc3.v0.2.8.PUBLIC.maf.gz in [https://gdc.cancer.gov/about-data/publications/mc3-2017]). Germline and somatic calls from PCAWG excluding ESAD-UK and MELA-AU were downloaded from the ICGC data portal [https://dcc.icgc.org/repositories]; these were available under restricted access, which can be applied for via the ICGC DACO [https://daco.icgc-argo.org/]. Bam files for tumor and normal samples from MELA-AU (dataset ID: EGAD00001003388) and ESAD-AU (dataset ID: EGAD00001003580) were from the European Genome–Phenome Archive ([https://ega-archive.org]); they are available under restricted access, which can be applied for via the ICGC DACO [https://daco.icgc-argo.org/]. Mitochondrial somatic mutation calls in PCAWG were downloaded from [https://ibl.mdanderson.org/tcma/mutation.html]. Hartwig somatic and germline variant calls were downloaded after acquiring restricted data access from the Hartwig Medical Foundation [https://www.hartwigmedicalfoundation.nl/en/], request number DR-069; requests can be submitted at [https://www.hartwigmedicalfoundation.nl/en/data/data-acces-request/]. Replication timing data from lymphoblastoid cell lines to calculate the replicative strand bias was downloaded from [http://mccarrolllab.org/resources/]. Processed genomic region densities of expression, histone mark H3K36me3, replication timing, CTCF/cohesin-binding sites, and DNase I hypersensitive sites were obtained by contacting authors of original publication [https://doi.org/10.1016/j.cell.2017.07.003]. Genomic regions for the CRG75 alignability track and blacklisted regions by Duke and DAC were obtained from the UCSC Genome Browser [https://genome.ucsc.edu/cgi-bin/hgTables]. GnomAD v2 allele frequencies and pext scores were obtained from the gnomAD browser [https://gnomad.broadinstitute.org/downloads]. Gene coordinates were obtained from the UCSC genome browser [https://genome.ucsc.edu/cgi-bin/hgTables]. Regions with high amount of linkage disequilibrium were downloaded from [https://github.com/meyer-lab-cshl/plinkQC/tree/master/inst/extdata]. Pre-computed SpliceAI scores were downloaded from Illumina Basespace [https://basespace.illumina.com/projects/66029966]. Download of SpliceAI scores are free, but require generation of a free account at Illumina Basespace. Complete list of tested 891 genes in Supplementary Dataset 1. Missense tolerance score annotations were obtained from [http://biosig.unimelb.edu.au/mtr-viewer/downloads] and constrained coding region annotations from [http://quinlanlab.org/blog/2018/12/20/constrained-coding-regions.html]. Interaction scores from STRING v11.5 were downloaded from [https://string-db.org/cgi/download?sessionId=bPz0GBvgDw3p] and scores from HumanNet v3 (HumanNet-FN) from [https://www.inetbio.org/humannet/download.php]. Source data are provided with this paper.

## Code availability

Code can be obtained from https://github.com/lehner-lab/RDGVassociation and data is visualized at https://mischanvp.shinyapps.io/rare_association_shiny/.

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

## Acknowledgements

M.V.P. was supported by a Spanish Ministry of Science and Innovation FPI Fellowship (ref.: PRE2018-084410). Work in the lab of B.L. is funded by European Research Council (ERC) Advanced (883742) and Consolidator (616434) grants, the Spanish Ministry of Science and Innovation (BFU2017-89488-P, EMBL Partnership, Severo Ochoa Centre of Excellence), the Bettencourt Schueller Foundation, the AXA Research Fund, Agencia de Gestio d'Ajuts Universitaris i de Recerca (AGAUR, 2017 SGR 1322), and the CERCA Program/Generalitat de Catalunya. Work in the lab of F.S. is funded by the ERC Starting Grant HYPER-INSIGHT (757700), the Horizon2020 RIA grant DECIDER (965193), Spanish Ministry of Science and Innovation grant REPAIRSCAPE (PID2020-118795GB-I00), the EMBO YIP program, the Severo Ochoa Centre of Excellence award to IRB Barcelona, and the CERCA Program/Generalitat de Catalunya.

## Author contributions

M.V.P. collected and curated the data, implemented software, performed all analyses, visualized the data, and interpreted the results. S.P. supervised the processing of common germline variants for the PCA analysis and the definition of rare germline variants. J.E.C. performed data retrieval and bioinformatics preprocessing of the germline data of a part of the genomic data analyzed (TCGA WES data sets BRCA, COAD, GBM, KICH, KIRC, KIRP, LIHC, PAAD, READ, STAD, THCA and UCEC). D.O.M. performed data retrieval and bioinformatics preprocessing of a part of the genomic data analyzed (ICGC WGS data sets ESAD-UK and MELA-AU). F.S. and B.L. conceptualized the analysis, devised the methodology, interpreted results and supervised the study. M.V.P., B.L. and F.S. drafted the manuscript jointly.

## Competing interests

The authors declare no competing interests.
