## [Peer review file · Nature Communications]

REVIEWER COMMENTS

Reviewer #1 (Remarks to the Author): Expert in computational cancer genomics and pathogenic germline variants in cancer

Pour et al. presents here a tour de force germline-somatic analyses focusing on rare damaging germline variants (RDGV) giving rise to somatic mutation signatures. In general, the analyses were carefully conducted and many factors considered. More statistical rigor on the germline sides is needed: (1) The power to identify new RDGV, esp outside of known genes, will be directly affected by the freq of the carriers and need to be transparent. (2) There are redundancies in how results are presented, and (3) associations need further validation if only found in small number of carriers. Overall, the reviewer finds the study to be carefully designed and almost did too much to be presented in one paper (ex. not even commenting on how autoencoder or ICA can identify new genome damaging signatures here).

Major comments:

1. Power determines results, and is directly affected by the study cohort's sample size & number of carriers of the germline gene. Given that the study already compiled some of the largest available cohorts & adding samples at this point only give diminishing returns, it'd be good to at least be very transparent about these numbers.

a. Figure 5 should be upfront and the number of carriers for each gene with association needs to be clear.

2. Redundancy in the 15 models, i.e., dominant/recessive/additive results are correlated, not to mention the RDGV sets are directly overlapping/containing each other.

- This statement is problematic and needs to be re-written. "In summary, with regards to the model of inheritance, RDGV set, and component (mutational process) extraction method, there was no single best model and most models added unique associations to the results." If you look at the overall numbers of course each model will add counts. But, if there is a biological ground truth, each individual association should be best captured by the model that describes that association's mode of inheritance, functional variants, etc. May be interesting to point out a few cases that the analyses clearly suggest a specific mode of inheritance rather than another one for a given association.

o Be good to describe in Results how additive/recessive models directly use LOH information, as this differ from typical germline analyses.

- In germline studies. it's custom to eventually pick one to do all subsequent analyses (typically additive since it can often capture associations from dominant/recessive). The reviewer understands the advantage and the authors' efforts of trying out different ones but it could be mis-leading. Perhaps after

describing results of testing all models, focus the presentation of association on one model that give the most best hits/replication rate etc.

- In Fig 3 (& S4), it's misleading to put in how many models as possible strength of association. To kill two-birds in one stone, the authors can consider replacing the number as the # of the observed RDGV in that cancer type (solving major comment 1), in a chosen inheritance model (maybe additive if want to be all encompassing).

3. Robustness of associations & biomedical impacts, can be challenging because aside from the apparent (but known) BRCA/MMR gene associations, the other ones probably need more evidence to be convincing. The authors already did a good job doing STRING-analyses and compiling literature.

a. Unlikely given it's rare variants, but good to make sure the claimed "novel" genes are not too close in LD (tag a specific haplotype) or overlap the known BRCA/MMR genes.

b. Any of these genes also showed association with cancer risk in literature?

Minor comments:

1. Scientists/journals are often intrigued to derive new terms for unnecessary novelty claims, which put burdens on the readers. In this case, rare damaging germline variants (RDGV) is similar to pLoF (predicted loss-of-function) or other terms more used by others in population germline studies. Can consider replacing it.

a. Whichever term use eventually, need to define what RDGV/pLoF means in the context of this paper, i.e., now on page 7 line 299 should be upfront in the section starting to describe germline data.

2. Only Additional Methods References but not References are in the combined manuscript file.

Reviewer #2 (Remarks to the Author): Expert in cancer genomics and germline variant pathogenesis

This manuscript by Vali Pour et al describes a systematic analysis of multiple large-scale WES and WGS datasets (TCGA, PCAWG, and Hartwig) to enumerate rare damaging germline variants (RDGVs) and identify associations with different somatic mutational processes. Multiple features of mutational processes were considered, and two different dimensionality reduction techniques (ICA and VAE) were used to remove redundancy and collapse onto 29 somatic mutation components. The authors were able to recapitulate several known associations as well as identify novel putative associations. They additionally used network analysis to support some of their novel findings. While the analysis is

interesting and produced many potentially novel hypotheses, there are several limitations that should be addressed prior to publication:

1. The paper is highly technical and complicated and at times hard to follow. The text should be simplified and edited for clarity. Some of the methodological details can be moved from the Results to Methods so that key findings and novel insights are more prominent in the narrative.

2. While ICA and VAE were used to reduce dimensionality of somatic signatures, from Figure 1d it seems like they identified similar components (e.g., dMMR_VAE1 and dMMR_ICA) and therefore re-introduced redundancy. The rationale for retaining both is unclear. Is an association with both dMMR_VAE1 and dMMR_ICA considered to be more reliable than association with one and not the other?

3. So many potential associations were tested involving different cancer types, genes, signatures, definitions of RDGVs, and modes of inheritance. Yet except for the associations between BRCA1/BRCA2/PALB2 and HRD-related signatures, most other detected associations were specific to unique combinations of definitions and models. The authors conclude that “with regards to the model of inheritance, RDGV set, and component (mutational process) extraction method, there was no single best model and most models added unique associations to the results”. I don’t believe this is well supported, as it seems like some of the 200+ identified associations are more reliable than others. Moreover, it would be interesting to explore why some models produced significant associations with certain signatures and others did not, and whether this yielded new biological or technical insights about different genes exhibiting different patterns.

4. Can the authors estimate the sensitivity for recovering known associations? False negatives seem to be a potential issue. Even with permissive thresholds, only 3 genes (BRCA1, EP300, MTOR) were associated with the same somatic mutational component across two different cancer types; this seems low. Per Figure 3, BRCA1 and BRCA2 were not associated with dHR in any single cancer type besides breast cancer. Lynch Syndrome genes MSH6 and PMS2 appear to not be associated with dMMR pan-cancer nor in any cancer type. Is there an explanation for why these associations may not have been detected?

5. Related to the above (i.e., the absences of associations with MSH6 and PMS2), what does that mean for the other genes associated with dMMR signatures, such as EXO1, TRAAP, SETD1A, TTI2, etc? What is one to make of the fact that EXO1 is associated with somatic mutational components dMMR_VAE2 and not dMMR_ICA and dMMR_VAE1, which is the opposite of MSH2 pan-cancer?

6. The analysis was restricted to the European population only. This is not well justified, nor is it clearly explained how the inclusion of other populations would have affected the results.

Reviewer #3 (Remarks to the Author): Expert in mutagenesis, evolution, and population genetics

This manuscript presents a novel approach to identify genes that influence somatic mutation processes by searching for association between somatic mutation patterns in tumor samples and rare germline variants carried by patients in three large-scale cohorts (TCGA, PCAWG and Hartwig). With this approach, the authors identified dozens of mutational components and 207 robust gene-mutational component associations involving 42 unique genes. It is reassuring that this analysis replicated many DNA repair genes (e.g., BRCA1, BRCA2, MSH2, MLH1) in which inherited deleterious variants are known to increase human mutation rates and predispose carriers to cancer. The identified associations also revealed some new genes whose roles in corresponding DNA repair pathways have been implicated in previous literature and/or are further supported by protein-protein interaction network analysis.

This paper utilizes cutting-edge statistical analysis to identify genes involved in somatic mutagenesis in humans and to uncover the genetic component of variability in mutation rates, thus making a timely contribution to this rapidly developing area of research. I find the methods innovative on three points: (1) inclusion of diverse mutation features in addition to the counts of different mutation types, including the regional distribution and strand asymmetry of mutations; (2) application of two dimensionality reduction methods to extract “mutational components” from complex and often correlated mutation features; (3) use of a combined burden and variance test (SKAT-O) under different inheritance models to allow for flexible model assumptions. However, these methodological innovations somewhat impair or limit interpretation of the results: I have several questions about the nature of the “mutational components” and interpretation of the associations detected. In addition, not much biological insights are generated other than the broad the conclusion that mutational processes in humans are affected by numerous genes. Lastly, it is highly unclear how the findings help understand and predict variation in mutation load and cancer risk across individuals. Given these limitations, I think this paper may be better suited for a journal focused on genetics or mutation research.

Major points:

1. Classic mutational signature analysis (e.g., Alexandrov et al 2013) utilizes non-negative matrix factorization (NMF) technique by treating the observed mutations as the combined results of many mutational processes each generate a specific combination of mutation types, which are defined by trinucleotide context only. The simplified definition of mutation types is a major weakness of this method, but the NMF model makes sense, as it is intuitive to assume that all mutational signatures have non-negative activities and lead to non-negative numbers of mutations of any type. One innovation point of this manuscript under review is the consideration of additional mutation features (e.g., regional distribution with regard to chromatin states), but the methods (ICA and VAE) the authors used to extract “independent mutational components” from these redundant features lost intuitive interpretation. What does this “independence” mean in statistical sense and in genetics context? What exactly is a “mutational component”? The authors seem to assume that each mutational component corresponds

to a mutational process or a form of repair deficiency without explanation or justification. Can one translate a “mutational component” into a “mutational signature” based on the loadings of all the features? Is there theoretical guarantee or empirical support that there won't be negative number of mutations associated with a mutational component? If there is inherent difficulty in interpreting results from ICA and VAE, is it possible to apply the NMF model to an extended panel mutation types defined with features in addition to trinucleotide context?

2. It is a great idea to apply the combined test instead of the burden test to detect associations, as it allows for the possibility that the rare variants have zero or opposite effects. However, I have a hard time understanding what type of effects these associations represent. Increasing or decreasing mutation rates? Shifting the distribution of mutations towards certain genomic regions (e.g., late versus early replicating regions)? I guess this question largely stems from the lack of biological interpretation of the “mutational components”. In addition, I'm surprised by the result that most association signals come from the variance test rather than the burden test even for protein truncating variants (Fig 2i), all of which presumably have the same loss-of-function effect. Is this due to imperfect variant annotation or other reasons? Further,

3. There is relatively weak validation for the identified associations. The authors presented three types of evidence: 1) statistical replication in a cohort (PCAWG + Hartwig) that is independent from the TCGA discovery cohort; 2) support from previous literatures; 3) enrichment of protein-protein interactions among genes involved in the associations. Evidence from 1) is not as strong as it appears, as the actual false discovery rate can be as high as 21% and 44% at the 1% and 2% “empirical FDR” levels, respectively; the FDR would be much higher for the newly identified genes as many associations involve well-known genes such as BRCA1 and MSH2. Evidence from 3) is also weak, as the significance is only marginal ($p > 0.03$) after removing five previously known genes (Fig 2c,d), and genes associated with the same mutational components do not appear to interact with each other more often (Fig 4e). Evidence 2) can be considered as sort of functional validation, but it is possible that many genes had been reported to be involved in DNA repair pathways before, so it is unclear how significant such literature support is. In general, I am not sure which of the novel associates are the most reliable and what biological insights are gained from this paper.

4. As the authors stated, one of the motivations for rare-variant association study is to “understand and predict variation in cancer risk among individuals” (page 3). However, the study as currently presented did not identify specific causal variants or estimate the effect sizes from the association results (especially the variance test), which prohibits explaining and predicting variation in mutation burden or cancer risk. This is a major limitation of the methods (both the dimensionality reduction method and the association method) used.

5. The authors argued for prevalence of genetic modifiers of somatic mutational processes by showing the aggregated frequency of rare variants in associated genes (Fig 5). However, this metric is likely an over-estimation for two reasons: (1) the TCGA cohort is a collection of tumor samples (and matching normal tissues) and is thus enriched for cancer predisposing variants than the general population, so it would be more relevant to calculate the frequency based on gnomAD or other datasets; (2) the small p parameters for most associations suggest many of the tested variants are not causal (or some have protective effects).

Minor points:

1. Only germline variants observed in gnomAD at allele frequency lower than 0.1% were considered for association; further, variants with $\geq 1\%$ frequency in the cohort were removed. What is the rationale for considering rare variants only? Is there any justification for the 0.1% and 1% thresholds? How would these thresholds affect the results?
2. In Fig 3 and Extended Data Fig 5, why did the authors order the columns (i.e., genes) by hierarchical clustering based on CRISPR-derived fitness scores? It seems more straightforward to order the genes by the mutational components that they are associated with, which will also facilitate reading of the text in the various section in pages 9-11. A side note: please label the colored genes within the figure so the readers can understand the colors without having to read the legend.
3. How to interpret a gene-mutational component association that is significant under multiple models of inheritance (for example, and the intersection area in the Venn diagram in Fig 2f and BRCA1/2 in Fig 3)? Can the authors use model competition to reduce the possibilities?
4. Association between SETD2 and MMR deficiency is highlighted in the abstract and discussed on page 10, but this association was detected at a relatively high FDR level (the actual FDR was estimated to be 44%) and not even shown in Fig 3. Please explain why this case is remarkable.
5. The manuscript is a bit long. I hope the authors can consider streamline the text to highlight the most important findings rather than to briefly touch on all “interesting” cases.
6. Page 4: “identifiy” should be “identify”; page 11: “an decrease” should be “a decrease”.

Reviewer #1 (Remarks to the Author): Expert in computational cancer genomics and pathogenic germline variants in cancer

Pour et al. presents here a tour de force germline-somatic analyses focusing on rare damaging germline variants (RDGV) giving rise to somatic mutation signatures. In general, the analyses were carefully conducted and many factors considered. More statistical rigor on the germline sides is needed: (1) The power to identify new RDGV, esp outside of known genes, will be directly affected by the freq of the carriers and need to be transparent. (2) There are redundancies in how results are presented, and (3) associations need further validation if only found in small number of carriers. Overall, the reviewer finds the study to be carefully designed and almost did too much to be presented in one paper (ex. not even commenting on how autoencoder or ICA can identify new genome damaging signatures here).

We thank the reviewer for the enthusiasm and for labelling the study a tour de force. Please find below the answers to the individual points raised.

Major comments:

1. Power determines results, and is directly affected by the study cohort's sample size & number of carriers of the germline gene. Given that the study already compiled some of the largest available cohorts & adding samples at this point only give diminishing returns, it'd be good to at least be very transparent about these numbers.

We thank the reviewer for these comments. In the revised manuscript we present two additional analyses: a power analysis and a subsampling of the validation cohort.

Power analysis in gene-based (burden) testing approaches is challenging due to the many parameters and the mixed, unknown effects from the different variants occurring in a gene. Thus, we utilised two different approaches.

Firstly, we employed an existing power analysis tool which was specifically designed for aggregate-level association tests, namely PAGEANT [PMID: PMC5925788]. We calculated the median power of each study depending upon the sample size and the pre-specified total variance explained (TVE) by all variants in a gene (related to effect size of the gene). As shown in Supplementary Figure 3, the estimated power for the individual cancer types was at a median of 22 % (1st quartile 13 %, 3rd quartile 29 %) when the amount of phenotypic TVE was 2%, and it was at a median of 7 % (1st quartile 4 %, 3rd quartile 10 %) when the TVE was 1 %. These two amounts of variance-explained correspond to genes known to affect mutation spectra in some tissues: *BRCA1* (2% variance) and *MLH1* (1 %) in our pan-cancer discovery

cohort (Supplementary Figure 4). We expect that the variance-explained for additional genes will likely be not higher than for these known genes, and so the power to identify these additional genes will be lower than for the examples given above. Supplementary Figure 5 shows the statistical power for the various % TVE, estimated for genomic datasets of different sizes.

Secondly, in addition to the power analysis above, we also performed a simple subsampling analysis, as shown in Supplementary Figure 6. In brief, we randomly subsampled the individuals from the validation cohort (in triplicate) retaining 95 %, 90% and 80 % of the individuals. The number of replicated associations dropped from 207 to 162-175 (at a FDR of 1 %) and from 356 to 286-311 (at a FDR of 2 %), when 80 % of the data was retained. Further, this analysis showed how even by removing 5 % of the data (around 230 samples), the number of replicated hits still decreased somewhat. This suggests that the replication rate is very sensitive to sample size and that a higher sample size will increase the validation rate.

In summary, both approaches suggest that, with the utilised sample sizes, low statistical power underlies the low validation rate and consequently that there is a high rate of false-negatives (i.e. non-replicated associations) in this study. We note that this does not in any way invalidate those associations that did replicate.

a. Figure 5 should be upfront and the number of carriers for each gene with association needs to be clear.

We thank the reviewer for this comment. We added the number of carriers for each replicated gene-cancer type pair in Figure 3, as well as to Supplementary Figure 14. In addition, we added Supplementary Figure 16 showing the distribution of rare variants across rare pLoF variant sets, and cohorts (discovery and validation). Finally we added Supplementary Figure 17 showing the total number of individuals carrying a rare pLoF variant in the 86 replicated genes (and in addition some known dMMR/dHR genes), across rare pLoF variant sets and cohorts.

2. Redundancy in the 15 models, i.e., dominant/recessive/additive results are correlated, not to mention the RDGV sets are directly overlapping/containing each other.

- This statement is problematic and needs to be re-written. "In summary, with regards to the model of inheritance, RDGV set, and component (mutational process) extraction method, there was no single best model and most models added unique associations to the results." If you look at the overall numbers of course each model will add counts. But, if there is a biological ground truth, each individual association

should be best captured by the model that describes that association's mode of inheritance, functional variants, etc. May be interesting to point out a few cases that the analyses clearly suggest a specific mode of inheritance rather than another one for a given association.

We thank the reviewer for the comment. We agree that each association should be best captured by the model that corresponds to the correct mechanism. In most cases however is difficult to determine this for each association, since (1) no effect size estimates are provided by SKAT-O and (2) it is difficult to rely on comparing p-values since the sample sizes differ between models e.g. the recessive model often had a low number of samples with biallelic inactivations available for testing. For instance, *BRCA1/2* associated with dHR phenotypes via all 3 models of inheritance, even though current literature suggests a recessive model [PMID: 31292550]. We could have not made this conclusion based on our data analysis without the additional literature support.

This conclusion is also supported by our subsampling analysis. In brief, we performed subsampling in triplicates by randomly retaining 95 %, 90 % and 80 % of the data of the validation cohort (in triplicate for each scenario). The number of replicated hits decreased with less data and, more importantly for the matter at hand, also varied within triplicates of the sampling (Supplementary Figure 6). As shown in Supplementary Figures 7-11 depending on the gene-cancer_type-phenotype triplet, some combinations did not associate with a specific model of inheritance-pLoF variant set anymore, depending on the amount of data retained in the random subsampling simulation. While *BRCA1* and *BRCA2*, for instance, still associated with all combinations even when only 80 % of the data was retained, this was not the case for *PALB2*, *MSH2*, and *MSH6* anymore. For instance, many replicated *MSH2* associations were not significant anymore when subsampling the data and the number of associations also varied between triplicates (Supplementary Figure 11). Thus, small variations in sample size and also random factors in sampling can impact the validation rate when a specific association is close to the p-value threshold, cautioning against directly comparing models with each other to determine which one is optimal. Thus, we removed the corresponding statement and added further clarification in the manuscript on how many associations were identified with a specific mode of inheritance (Figure 3 and Supplementary Figure 13), and the motivation for this approach (see Discussion 1st paragraph).

o Be good to describe in Results how additive/recessive models directly use LOH information, as this differ from typical germline analyses.

Thank you for the comment. We changed the text accordingly (in the Results subsection, see "Rare variant association with a combined burden and variance test" 1st paragraph).

- In germline studies. it's custom to eventually pick one to do all subsequent analyses (typically additive since it can often capture associations from dominant/recessive). The reviewer understands the advantage and the authors' efforts of trying out different ones but it could be mis-leading. Perhaps after describing results of testing all models, focus the presentation of association on one model that give the most best hits/replication rate etc.

We thank the reviewer for this comment. Indeed, in the past most studies focussed on one model (often additive) and one pLoF variant set. This way, however, it was not clear how other models could have changed the results. With this in mind, we decided to test several models of inheritance and pLoF variant sets, since we did not know *a priori* which combination would best capture the biological signal. In addition, it is expected that the best model may depend on the genes. This approach was inspired by a recent large-scale study in which rare variants from 281,104 UK Biobank exomes were analysed using 11 different models (10 dominant, 1 recessive) [PMID: 34375979]. We agree that interpretation of the results can be more challenging with this multiplicity of models, and thus we added further explanation and more detailed visualisation in the manuscript to guide the reader (see Figure 3, Supplementary Figures 13 &14, Discussion new text at the end of the first paragraph).

- In Fig 3 (& S4), it's misleading to put in how many models as possible strength of association. To kill two-birds in one stone, the authors can consider replacing the number as the # of the observed RDGV in that cancer type (solving major comment 1), in a chosen inheritance model (maybe additive if want to be all encompassing). **We thank the reviewer for these helpful comments. We changed Figure 3 and Supplementary Figure 13-14 (former S4) accordingly to make it clear via which model (and via which rare pLoF variant set) each replicated gene was identified. We also displayed the number of rare pLoF variants (+ LOH). This way, we provide an overall view of all the results, and 'zoom into' individual cases in the text.**

3. Robustness of associations & biomedical impacts, can be challenging because aside from the apparent (but known) BRCA/MMR gene associations, the other ones probably need more evidence to be convincing. The authors already did a good job doing STRING-analyses and compiling literature.

a. Unlikely given it's rare variants, but good to make sure the claimed "novel" genes are not too close in LD (tag a specific haplotype) or overlap the known BRCA/MMR genes.

Indeed this would be an interesting analysis to perform, but it is challenging to estimate LD for rare variants. As a related analysis, we measured the genomic distances between our newly replicated genes and known dMMR/dHR genes, operating under the reasonable assumption that the distance is well correlated

with LD. We generated a table (Supplementary Table 1) capturing for each gene the closest gene of the known causal genes (dMMR/dHR). For 50 out of 81 newly replicated genes (at a FDR of 2 %) no other dMMR/dHR gene was detected on the same chromosome. For the remaining genes, the closest distance to a known dMMR/dHR gene was at a median of 35.2 MB (1st quartile: 9.1 MB, 3rd quartile: 101.4 MB; Supplementary Table 1). LD to a known gene is therefore very unlikely to underlie our associations.

b. Any of these genes also showed association with cancer risk in literature?

We thank the reviewer for the comment. Indeed variants in *MSH3* have been associated with an increased cancer risk [PMID: 23946381]. Out of the hits at a FDR of 1%, 7 genes are known cancer predisposition genes [PMID: 24429628] (*BRCA1*, *BRCA2*, *FANCC*, *MLH1*, *MSH2*, *PALB2*, and *APC*) and out of the hits at a FDR of 2%, 6 additional cancer predisposition genes [PMID: 24429628] were identified (*AXIN2*, *COL7A1*, *DIS3L2*, *DOCK8*, *SOS1*, and *WRN*). We've now added this information to the manuscript (in the Results subsection "42 genes robustly associated with somatic mutation phenotypes" end of 2nd paragraph).

Minor comments:

1. Scientists/journals are often intrigued to derive new terms for unnecessary novelty claims, which put burdens on the readers. In this case, rare damaging germline variants (RDGV) is similar to pLoF (predicted loss-of-function) or other terms more used by others in population germline studies. Can consider replacing it.

We thank the reviewer for the comment and we have changed the text and figures accordingly -- we now use the "pLoF" term throughout the manuscript.

a. Whichever term use eventually, need to define what RDGV/pLoF means in the context of this paper, i.e., now on page 7 line 299 should be upfront in the section starting to describe germline data.

Thank you for the comment. We changed the text accordingly.

2. Only Additional Methods References but not References are in the combined manuscript file.

We thank the reviewer for pointing this out.

Reviewer #2 (Remarks to the Author): Expert in cancer genomics and germline variant pathogenesis

This manuscript by Vali Pour et al describes a systematic analysis of multiple large-scale WES and WGS datasets (TCGA, PCAWG, and Hartwig) to enumerate rare damaging germline variants (RGDVs) and identify associations with different somatic mutational processes. Multiple features of mutational processes were considered, and two different dimensionality reduction techniques (ICA and VAE) were used to remove redundancy and collapse onto 29 somatic mutation components. The authors were able to recapitulate several known associations as well as identify novel putative associations. They additionally used network analysis to support some of their novel findings. While the analysis is interesting and produced many potentially novel hypotheses, there are several limitations that should be addressed prior to publication:

We thank the reviewer for the enthusiasm and suggestions for additional analyses that we detail below.

1. The paper is highly technical and complicated and at times hard to follow. The text should be simplified and edited for clarity. Some of the methodological details can be moved from the Results to Methods so that key findings and novel insights are more prominent in the narrative.

Thank you for the comment. We changed different parts of the manuscript to clarify our approach and to focus on the novel findings, such as (i) clarifying the different utilised models (see Results subsection “Rare variant association with a combined burden and variance test”), (ii) shortening the section in the main text explaining SKAT and moving it to the methods, (iii) moving the inflation analysis description to the supplementary text, and (iv) moving the technical explanation of the calculation of the FDRs to the methods while keeping the idea behind it. All changes are highlighted in the revised text.

2. While ICA and VAE were used to reduce dimensionality of somatic signatures, from Figure 1d it seems like they identified similar components (e.g., dMMR_VAE1 and dMMR_ICA) and therefore re-introduced redundancy. The rationale for retaining both is unclear. Is an association with both dMMR_VAE1 and dMMR_ICA considered to be more reliable than association with one and not the other?

Thank you for the comment. To reduce the multiple testing burden, and for ease-of-interpretation, we wanted to remove redundancies/correlations between the 56 input features describing mutation patterns. Inspired by a previous study on gene expression data [PMID: 32393369] that showed that different dimensionality reduction techniques could capture different biological processes, we decided on using two different techniques in parallel: ICA and VAE. Since we did not know *a priori* which components would work

better (e.g. would dMMR_VAE1 and dMMR_ICA associate with the same genes?), we tested the components derived from both VAE and ICA, which indeed may have re-introduced some redundancy. While some associations were identified with components from both tools, others were exclusively identified via one method, which we think would argue for retaining components from both methods for association testing, while being transparent about possible re-introduced redundancy. We now make this explicit in the text (added in Results in subsection “Somatic mutation phenotypes in 15,000 human tumors” last paragraph. For instance, as seen in Fig. 3 *BRCA2* in breast cancer associated with dHR_{ICA}, but not with dHR_{VAE2}, and similarly it can be seen that some genes associated with dMMR_{VAE1}, but not with dMMR_{ICA} such as *KANSL1* in lung. We do not think that an association with both dMMR_VAE1 and dMMR_ICA -- or other such pairs of ICA and VAE components -- would necessarily be considered more reliable than an association with just one member of the pair.

3. So many potential associations were tested involving different cancer types, genes, signatures, definitions of RDGVs, and modes of inheritance. Yet except for the associations between *BRCA1/BRCA2/PALB2* and HRD-related signatures, most other detected associations were specific to unique combinations of definitions and models. The authors conclude that “with regards to the model of inheritance, RDGV set, and component (mutational process) extraction method, there was no single best model and most models added unique associations to the results”. I don’t believe this is well supported, as it seems like some of the 200+ identified associations are more reliable than others. Moreover, it would be interesting to explore why some models produced significant associations with certain signatures and others did not, and whether this yielded new biological or technical insights about different genes exhibiting different patterns.

We thank the reviewer for the comment. We removed the corresponding statement and would like to point to our response to a similar comment by reviewer 1 (see response to point 2 of reviewer 1). In short, we do not believe that it is helpful to compare between models-of-inheritance in this analysis due to the low power and due to different sample sizes between models-of-inheritance, as further illustrated by our subsampling analysis (new Supplementary Figure 6-11).

4. Can the authors estimate the sensitivity for recovering known associations? False negatives seem to be a potential issue. Even with permissive thresholds, only 3 genes (*BRCA1*, *EP300*, *MTOR*) were associated with the same somatic mutational component across two different cancer types; this seems low.

Per Figure 3, *BRCA1* and *BRCA2* were not associated with dHR in any single cancer type besides breast cancer.

***BRCA1/BRCA2* associations with dHR phenotypes were indeed identified in the discovery cohort at a FDR of 1% also in other cancer types (e.g. ovarian cancer, stomach + oesophagus cancer). In total, 54 associations between *BRCA1/2* and dHR phenotypes in cancer types other than breast and pan-cancer were identified at a FDR of 1%, involving 7 different cancer types. Out of these 54, 26 could be re-tested in the validation cohort (based on the available number of variant-bearing individuals), but none of them reached significance after multiple testing corrections. As described in reviewer response point 1, statistical power was decreased in the individual cancer types leading to high false-negative rate in the validation set (Supplementary Figures 3 & 4). For instance, we estimated for *BRCA2* a lower-bound of total variance explained of 5%, which would result in a statistical power of 64% in the discovery cohort and of 62% in the validation cohort in ovarian cancer. For *BRCA1*, the total variance explained was around 2%, leading to lower statistical power. We have added a comment on this in the Discussion of the revised manuscript (see Discussion 5th paragraph).**

Lynch Syndrome genes *MSH6* and *PMS2* appear to not be associated with dMMR pan-cancer nor in any cancer type. Is there an explanation for why these associations may not have been detected?

We thank the reviewer for this comment, and indeed one would have expected associations between these genes with dMMR phenotypes. We identified in the discovery cohort 4 *MSH6* associations in prostate cancer with dMMR phenotypes and 2 *PMS2* associations in kidney cancer with dMMR phenotypes. However, only the *MSH6* associations could be re-tested in the validation cohort (due to requirements on the number of variant-bearing individuals per cancer type), where they did not reach significance. This again illustrates that larger sample sizes should allow replication of many more associations (as supported by our subsampling analysis; see Response to reviewer 1 point 1 & 2). Further, as described above and as shown in Supplementary Figures 3 and 4, the “total variance explained” metric (see Response to reviewer 1 point 1) by *MSH6* and *PMS2* was at around 1% and 0.4 % respectively (lower bound estimate), leading to a lower power. This may explain why these associations were not identified at this sample size in comparison to other high effect size genes such as *BRCA2* (~5 % of variance explained) and *MSH2* (~4% of variance explained).

5. Related to the above (i.e., the absences of associations with *MSH6* and *PMS2*), what does that mean for the other genes associated with dMMR signatures, such as *EXO1*, *TRAAP*, *SETD1A*, *TTI2*, etc? What is one to make of the fact that *EXO1* is associated with somatic mutational components dMMR_VAE2 and not dMMR_ICA and dMMR_VAE1, which is the opposite of *MSH2* pan-cancer?

We thank the reviewer for this comment. As indicated in the added Supplementary Figures 3 & 4, statistical power in this study was modest,

which makes identifying the associations more susceptible to noise. This was also shown in our subsampling analysis (Supplementary Figures 6-11), which illustrates how low variations in the sample size can ‘push’ an association over the significance threshold or the opposite, leading to a high fraction of false-negatives in the replication analysis. This could also explain why some associations only replicated with a specific phenotype, e.g. see *MSH2* in Supplementary Figure 11.

6. The analysis was restricted to the European population only. This is not well justified, nor is it clearly explained how the inclusion of other populations would have affected the results.

We thank the reviewer for this comment. We restricted our analysis to a specific ancestry to exclude the possibility of population-specific germline variants showing spurious associations with mutational processes (since mutational processes might differ between populations also due to different external exposures, but would be apparently associated with population-specific variants; here it would be hard to distinguish confounding from real causation, e.g. as in the classical example of the *SUSH1* gene [Hamer and Sirota; *Molecular Psychiatry* (2000) 5, 11–13; PMID: 10673763]). We selected samples of European ancestry, since this was the largest group in our cohort (e.g. in TCGA-Discovery ~7,000 Europeans, ~1,000 Africans, and ~500 Asians). We do agree that investigating the effects of germline variants on mutational processes in other populations would be of interest in future studies that will draw upon larger sample sizes.

Reviewer #3 (Remarks to the Author): Expert in mutagenesis, evolution, and population genetics

This manuscript presents a novel approach to identify genes that influence somatic mutation processes by searching for association between somatic mutation patterns in tumor samples and rare germline variants carried by patients in three large-scale cohorts (TCGA, PCAWG and Hartwig). With this approach, the authors identified dozens of mutational components and 207 robust gene-mutational component associations involving 42 unique genes. It is reassuring that this analysis replicated many DNA repair genes (e.g., BRCA1, BRCA2, MSH2, MLH1) in which inherited deleterious variants are known to increase human mutation rates and predispose carriers to cancer. The identified associations also revealed some new genes whose roles in corresponding DNA repair pathways have been implicated in previous literature and/or are further supported by protein-protein interaction network analysis. This paper utilizes cutting-edge statistical analysis to identify genes involved in somatic mutagenesis in humans and to uncover the genetic component of variability in mutation rates, thus making a timely contribution to this rapidly developing area of research. I find the methods innovative on three points: (1) inclusion of diverse mutation features in addition to the counts of different mutation types, including the regional distribution and strand asymmetry of mutations; (2) application of two dimensionality reduction methods to extract “mutational components” from complex and often correlated mutation features; (3) use of a combined burden and variance test (SKAT-O) under different inheritance models to allow for flexible model assumptions. However, these methodological innovations somewhat impair or limit interpretation of the results: I have several questions about the nature of the “mutational components” and interpretation of the associations detected. In addition, not much biological insights are generated other than the broad the conclusion that mutational processes in humans are affected by numerous genes. Lastly, it is highly unclear how the findings help understand and predict variation in mutation load and cancer risk across individuals. Given these limitations, I think this paper may be better suited for a journal focused on genetics or mutation research.

We thank the referee for the in-depth evaluation and for highlighting the various innovative aspects of the methods utilised, as well as the timeliness of the study. We do agree that the use of these various cutting-edge methods may present a challenge in interpretation of the results; please see our point-by-point responses below that attempt to address that with additional analyses and/or discussion.

Major points:

1. Classic mutational signature analysis (e.g., Alexandrov et al 2013) utilizes non-negative matrix factorization (NMF) technique by treating the observed mutations as the combined results of many mutational processes each generate a specific combination of mutation types, which are defined by trinucleotide context only. The simplified definition of mutation types is a major weakness of this method, but the

NMF model makes sense, as it is intuitive to assume that all mutational signatures have non-negative activities and lead to non-negative numbers of mutations of any type. One innovation point of this manuscript under review is the consideration of additional mutation features (e.g., regional distribution with regard to chromatin states), but the methods (ICA and VAE) the authors used to extract “independent mutational components” from these redundant features lost intuitive interpretation. What does this “independence” mean in statistical sense and in genetics context? **We thank the reviewer for highlighting the innovation of our method, in including diverse mutation-distribution features in our mutational phenotypes (in addition to the more standard trinucleotide SNV mutation frequencies).**

The reviewer touches upon several interesting and relevant issues here, which warrant further discussion. For instance, regarding the common use of NMF in analysing mutation patterns (which is just one of a myriad factorization techniques: reviewed and systematised e.g. at the website “Matrix Factorization Jungle” <https://sites.google.com/site/igorcarron2/matrixfactorizations>). One attractive property of NMF is that for many types of data it often yields ‘parts-based representations’ (that are additive and sparse) in practice; see Lee and Seung 1999.

Whether it makes intuitive sense that mutational processes would have nonnegative contributions: we would say this is often the case, but not always. E.g. active DNA mismatch repair (MMR) has a negative contribution to mutation rates. Then, inactivated MMR would register as a positive contribution and could be seen in NMF. However that NMF component does not describe MMR specificity faithfully: it is a superposition of the replication errors with relative efficiency of MMR on different types of errors. One could argue that here a factorization method that does allow negative coefficients would make more sense.

It is known that in practice NMF does have problems in disentangling certain mutational processes: for example, the known (Cosmic) signatures SBS3, SBS8 and SBS5 are routinely confounded in analyses (see e.g. simulation in [PMID: 34316057]). Regarding MMR failure signatures, 7 have been reported via NMF inference of tumor genomes [PMID: 32025018], however analysis of genomes in cell line models of MMR failure suggests that there would likely be only 2 or 3 MMR signatures. Overall we think NMF is only one of the possibilities to be applied for this sort of analysis. We’d like to note that NMF is also not convenient for analysing our mutation features with regional mutation rate enrichment (in chromatin states etc), since these features can be negative (log coefficients from the regression) and converting them to positive values is non-trivial and may involve a loss-of-information.

In terms of our use of ICA to extract “independent mutational components”: ICA is a methodology that seeks to maximise the statistical independence of the components. “Independence” is considered in a general sense in the ICA methods, and is not limited to e.g. linear Pearson correlation/covariance that is minimised by PCA. Ideally, also other forms of dependency between variables would be minimised including nonlinear correlations. The implementation of ICA that we (and many others) use, the “fastICA”, aims to maximise the non-Gaussianity of components (as a proxy for their statistical independence). This tends to work very well for “blind source separation” problems, such as unmixing of sound from multiple sources, or unmixing of overlaid images, where data is expected to be non-normally distributed in each channel. Our analysis is based on the intuition that mutation process data will also similarly fit this general type of distributions, being non-Gaussian across individuals e.g. a bimodal distribution of microhomology-flanked deletion burden resulting from individuals with active HR versus individuals with disabled HR. We have added a brief discussion of these issues in the Methods section of the manuscript, under the heading “Independent component analysis”.

What exactly is a “mutational component”? The authors seem to assume that each mutational component corresponds to a mutational process or a form of repair deficiency without explanation or justification. Can one translate a “mutational component” into a “mutational signature” based on the loadings of all the features? Is there theoretical guarantee or empirical support that there won't be negative number of mutations associated with a mutational component? If there is inherent difficulty in interpreting results from ICA and VAE, is it possible to apply the NMF model to an extended panel mutation types defined with features in addition to trinucleotide context?

We assume that our ICA/VAE mutational components correspond to mutational processes to the same degree that previous work employing NMF assumed that the NMF components corresponded to mutational processes.

Importantly, this assumption is conservative: if some of the components represent a mix of processes, or if they are a partial representation of a process, then associations with causal germline variants will have less power to be discovered and/or replicated. Therefore it is likely that the components that did replicate indeed largely correspond to biological processes.

We do not mean to imply that our mutational components can ‘translate’ one-to-one into existing, NMF-based mutational signatures. Indeed as the reviewer writes we did not intend to prevent that a component associates with a negative number of mutations. If it did, we think this is not necessarily a problem (see discussion above regarding DNA repair genes, whose activity prevents mutations). We do not think that this property prevents interpretation of the mutational components.

To further help interpretation, we now perform an additional analysis to ‘orient’ each association with respect to whether the variant, overall, associates with an increase or with a decrease in the contribution (‘exposure’) of each mutational component. Please see answer to Q2 below for details.

2. It is a great idea to apply the combined test instead of the burden test to detect associations, as it allows for the possibility that the rare variants have zero or opposite effects. However, I have a hard time understanding what type of effects these associations represent. Increasing or decreasing mutation rates? Shifting the distribution of mutations towards certain genomic regions (e.g., late versus early replicating regions)? I guess this question largely stems from the lack of biological interpretation of the “mutational components”.

We fully agree with this point -- it would certainly be helpful to have an idea of the direction of effects associated with deleterious variants in certain genes. This would help interpret biologically our mutational components. We have thus performed an additional analysis to ‘orient’ these effects.

The variance-test in SKAT-O does not in itself report the direction of associations (i.e. increase or decrease of effects linked with variants in a particular gene), since it was designed to be able to find bi-directional effects.

Operating under the assumption that most effects in our data (truncating or high-CADD variants affecting coding sequences) will be either LoF or no effect, but very rarely GoF, we can apply a simple test to assign a direction to the associations found significant by SKAT-O. For this purpose, we have run association testing with the same models using a classic burden test *via* a linear regression as explained in the methods (see Methods: Estimation of Effect Sizes via Burden Testing). The effect size estimate from this analysis can provide the direction for each association. We have added this information to Figure 3 and Supplementary Figure 13.

In addition, I’m surprised by the result that most association signals come from the variance test rather than the burden test even for protein truncating variants (Fig 2i), all of which presumably have the same loss-of-function effect. Is this due to imperfect variant annotation or other reasons? Further,

Indeed most validated associations derive from the variance-test, and we would also agree with the reviewer that the majority of variants (either truncating or missense) have LoF effects, rather than GoF effects. Then, as the reviewer writes, in the original manuscript we proposed that the reason that the variance-test produces more hits than the burden-test is because many of the missense variants prioritised by CADD have no or weak effect, despite a high CADD score.

It is indeed well-appreciated in the literature that algorithmic variant annotation is imperfect. It can also be seen in Figure 2i how applying a more stringent CADD threshold shifts the preference of the SKAT-O method (distribution of the rho parameter) towards the burden test and away from the variance test, consistent with enriching for causal missense variants with more stringent CADD threshold. As highlighted in the Discussion, we think that the use of variance-tests is an effective workaround to compensate for the inaccuracies in current tools for variant pathogenicity prediction. While variance-tests were designed to deal with heterogeneity of effects due to *biological* differences between variants, we think variance-tests are also a useful approach for dealing with the heterogeneity stemming from *technical* reasons i.e. imperfect accuracy of prediction of variant effects. Even with PTVs, the highest fraction of hits still replicated via SKAT and not burden (though clearly shifted more towards the burden-test, compared with missense variants; see Figure 3i). Again, not all PTVs automatically lead to a strong LoF, e.g. when located towards the end of a protein thus not triggering nonsense-mediated mRNA decay (NMD; Lindeboom et al. Nature Genetics 2019 PMID: 31659324).

3. There is relatively weak validation for the identified associations. The authors presented three types of evidence: 1) statistical replication in a cohort (PCAWG + Hartwig) that is independent from the TCGA discovery cohort; 2) support from previous literatures; 3) enrichment of protein-protein interactions among genes involved in the associations. Evidence from 1) is not as strong as it appears, as the actual false discovery rate can be as high as 21% and 44% at the 1% and 2% “empirical FDR” levels, respectively; the FDR would be much higher for the newly identified genes as many associations involve well-known genes such as BRCA1 and MSH2.

We are a bit surprised that the reviewer considers our validation using method 1) to be “relatively weak”. Replication of associations in an independent cohort is a standard way of demonstrating validity of associations.

The replication test is conservative, because of the differences between cohorts in sequencing techniques (WES vs WGS). This means that the mutational phenotypes will be estimated with varying precision (Supplementary Figure 44) across the two cohorts, and that the replication will be harder to achieve, thus meaning a conservative test.

Regarding “actual false discovery rate can be as high as 21% and 44% at the 1% and 2% “empirical FDR” levels, respectively”: we think that it is a strength of our study that we use an additional control, the random gene-sets, to estimate FDR. This is somewhat analogous to the use of e.g. synonymous mutations baseline in a recent multi-phenotype rare variant association study in Nature (PMID: 34375979), only in our case it is more conservative. This is

because the random gene-sets might plausibly contain genes that change mutation rates (yet-unknown DNA repair genes etc). Currently it is not possible to estimate how many of these there are, but there are likely to be some and so the 21% FDR estimate is a conservative upper bound of FDR for these associations.

Overall, we would not like to be penalised because of including additional controls in our study. We do agree with what (we understand) is the general intention underlying the reviewer's query: that additional analysis is needed to help prioritise and assign confidence to individual hits. Therefore we performed an additional randomization analysis based on network data; please see the second part of our answer to Q3, directly below.

Evidence from 3) is also weak, as the significance is only marginal ($p > 0.03$) after removing five previously known genes (Fig 2c,d), and genes associated with the same mutational components do not appear to interact with each other more often (Fig 4e). Evidence 2) can be considered as sort of functional validation, but it is possible that many genes had been reported to be involved in DNA repair pathways before, so it is unclear how significant such literature support is. In general, I am not sure which of the novel associates are the most reliable and what biological insights are gained from this paper.

To further prioritise individual hits resulting from our analysis and add confidence to their reliability, we capitalise on the network analysis we implemented (which is a methodological innovation for the rare-variant analyses). In particular, the current randomization test, using the network of protein-protein interactions (PPI), provides a p-value for the entire set of discovered genes, considered jointly. This p-value was, as the reviewer notes, significant also when removing all known (dMMR/dHR) genes.

We have now implemented two additional network-based tests on our 2% gene sets, with the goal of prioritizing individual hits. To this end we employed two different criteria, prioritizing hits by (i) their functional relatedness with the known germline mutation risk genes (dMMR and dHR) (Supplementary Figure 20), and by (ii) the strength of their interaction with their direct neighbours within the set of hits (Supplementary Figure 21).

In the first additional analysis, genes were prioritised based on their strength of interaction with known dMMR/dHR genes (see Methods and Supplementary Figure 20) with the idea that novel genes functionally related with one of these two pathways would likely have an effect on mutation rates. For this purpose, the strongest interaction between the tested (hit) protein and a dHR/dMMR protein was compared against the strongest interaction of other proteins with

dHR/dMMR proteins from the full list of 891 considered genes in our study, which had a similar node connectivity as the tested gene.

For the second additional analysis, genes were prioritised by their strength of interaction with their neighbours within the set of identified hits (associated with mutation processes but not dMMR/dHR genes). The underlying idea that interactions between new hits which are stronger than one would get from a random network, are more likely to be informative (see Methods and Supplementary Figure 21). In this analysis, random networks with the same number of genes from the full list of 891 considered genes were generated, while retaining the node connectivity. In this way, the extracted strongest interaction of a protein with its neighbour was compared against extracted interactions from 10,000 randomly shuffled networks.

In both analyses, by comparing an interaction score against a randomized baseline distribution, an empirical p-value was determined, and an effect size was calculated by subtracting the median of the baseline scores from the observed value of the interaction score.

We hope this additional approach provides a useful prioritization of the identified mutation process-associated genes, based on their functional consistency as a set. We note this methodology provides an orthogonal criterion to the FDRs from the genomic association analysis (which in themselves can be used to prioritise hits).

For example, some of the new hits that have high scores in their interaction with known dMMR/dHR genes are *RBBP8*, *MSH3*, *RAD51*, *MLH3*, *TP53BP1*, *EXO1*, and *WRN*, suggesting a higher confidence for these hits based on functional interaction data (Supplementary Figure 20).

We additionally hope that by implementing these tests we have strengthened the methodological innovation of our study, which could be used for other rare variant association studies to prioritise hits with the variety of phenotypes that are addressed in such studies.

4. As the authors stated, one of the motivations for rare-variant association study is to “understand and predict variation in cancer risk among individuals” (page 3). However, the study as currently presented did not identify specific causal variants or estimate the effect sizes from the association results (especially the variance test), which prohibits explaining and predicting variation in mutation burden or cancer risk. This is a major limitation of the methods (both the dimensionality reduction method and the association method) used.

We understand the general motivation underlying this remark: that associations with rare variants are interesting not only for mechanistic insight, but also for making predictive models e.g. for cancer risk. The latter are, in our opinion, outside the scope of the current study. How to combine different rare-variants into genome-wide risk scores is a matter of investigation and the first methods to do so are currently being developed (see e.g. PMID: 34615865 published approximately 5 months ago).

We hope that our associations will be useful (also) to generate risk scores in future work. The two additional analyses that we introduced in the revised manuscript -- as described in the answers to points 2 and points 5 -- should facilitate this uptake of our findings to the field.

To briefly recap here, these additional analysis that we think address this question are:

- (from Q2) The directionality for individual associations (are the rare damaging variants in gene X associated with a higher, or with a lower 'exposure' of mutational component Y)? In addition, our conservative estimates of the effect sizes (via the burden test) for each association have been made available.
- (from Q5) A comparison of prevalence of deleterious variants in genes-of-interest (association hits) in the cancer cohorts analyzed, versus healthy cohorts from gnomAD. Please see answers to Q2 and Q5 for details.

5. The authors argued for prevalence of genetic modifiers of somatic mutational processes by showing the aggregated frequency of rare variants in associated genes (Fig 5). However, this metric is likely an over-estimation for two reasons: (1) the TCGA cohort is a collection of tumor samples (and matching normal tissues) and is thus enriched for cancer predisposing variants than the general population, so it would be more relevant to calculate the frequency based on gnomAD or other datasets; (2) the small p parameters for most associations suggest many of the tested variants are not causal (or some have protective effects).

We agree with the referee that the population prevalence of the causal genetic modifiers of mutational processes should be better estimated, since this quantity is important to demonstrate the relevance of these variants to many individuals, and also to estimate the potential impact on cancer risk.

To address the possibility that the naive population-prevalence metric (in the TCGA cohort) is an overestimation due to the reasons that the reviewer mentions, in our original analysis we used control sets of randomly chosen genes, whose lengths were matched to the genes-of-interest (i.e. genes in which variants were associated with mutational phenotypes). Fig. 5 shows that these control genes have a lower prevalence of rare pLoF variants in the

cancer cohort, than the genes-of-interest in the same cancer cohort. This excess-prevalence of variants in the genes-of-interest over the control genes provides an estimate of the number of variants that may have a causal effect. As seen in Fig. 5, this excess-prevalence metric was broadly similar or higher for our FDR 1% hit genes to that for the known dMMR genes or the known dHR genes, supporting that some variants in the genes we identified are in fact cancer risk variants. This normalisation to length-matched genes should control for the fact that many of the rare variants deemed as pLoF are probably not causal (as evidenced in the small rho parameter of the SKAT-O test, as the reviewer notes).

In the revised manuscript, we have included an additional analysis to further address the reviewer's query: to consider germline variants in the same genes, however in non-cancer cohorts of individuals. This case-control comparison should provide an estimate about putative cancer risk associated with these variants; the results have been added to Figure 5. This analysis however has a caveat in that the case cohorts (TCGA, PCAWG, Hartwig) and the control cohort (gnomAD) were sequenced by different consortia and were processed using different bioinformatics pipelines, and so it is not possible to rule out differences due to technical biases. Of note, gnomAD does not provide individual-level data, but only population MAFs; therefore our analysis (out of necessity) was simply adding up the MAF of the variants that meet our RDGV criteria and are in the genes-of-interest. This is based on the reasonable (but sometimes violated) assumption that one individual has at most one rare pLoF variant in the set of genes-of-interest. Of note: this assumption is conservative, if it is violated, the estimation of cancer risk contributed by a gene will appear downward-biased). Further, we only used variants from the cancer genome sets which were also identified in gnomAD (excluding ultra-rare variants which were found in the cancer genomes but not gnomAD with high confidence) to make a more balanced comparison. We also consider the control gene set in this analysis, similarly as above.

In addition, we also compared the frequency of rare pLoF variants in gnomAD versus the cancer datasets, at the individual gene level for each of our replicated genes; it should be noted that this comparison was performed on very sparse data and so estimates for individual genes may not be highly precise (Supplementary Figure 18). For many of the genes we associated to various mutation patterns, the frequency of rare pLoF variants was increased in comparison to the healthy cohorts.

In summary, this new analysis shows that by comparing TCGA/Hartwig/PCAWG versus gnomAD, that the genes-of-interest have a higher frequency of pLoF variants in the cancer patient datasets compared to non-cancer-patient cohorts from gnomAD providing additional evidence that

RDGVs in these new genes, considered collectively, contribute to cancer risk. This analysis is summarised in the new panels in Figure 5, Supplementary Figure 18.

Minor points:

1. Only germline variants observed in gnomAD at allele frequency lower than 0.1% were considered for association; further, variants with $\geq 1\%$ frequency in the cohort were removed.

- What is the rationale for considering rare variants only? Is there any justification for the 0.1% and 1% thresholds?

The gnomAD minor allele frequency (MAF) threshold of 0.1% was inspired by previous studies, which applied similar thresholds, based on the understanding that common variants are usually not disease-causal variants. In the rare variant association studies in PCAWG [PMID: 32025007] and by Wang *et al.* [PMID: 31173346], a threshold of 0.5% was applied. In the recent UK biobank study [PMID: 34375979] depending on the collapsing model, different thresholds were used but for most models (4 out of 12) a threshold of 0.01% was used.

The threshold of rejecting $>1\%$ on the cohort was applied since we observed in a previous study from our labs [PMID: 29973584] that most variants with a frequency of $>1\%$ within the cohort, but that were however rare or not found in an external database (here gnomAD), were enriched with alignment/variant calling artefacts.

- How would these thresholds affect the results?

Applying a more stringent gnomAD MAF threshold would further enrich the variant set for damaging variants, but would also limit power to find associations since the variant set would be smaller. Relaxing the gnomAD MAF threshold would lead to a bigger variant set but also increase the amount of non-causal variants in the dataset. We take the opportunity to highlight that our use of MAF thresholds is not the only method we use to ascertain deleteriousness of variants: there is also the CADD threshold (which we did vary, and we did consider the effects of CADD threshold on association calls, as shown in Figure 2i).

2. In Fig 3 and Extended Data Fig 5, why did the authors order the columns (i.e., genes) by hierarchical clustering based on CRISPR-derived fitness scores? It seems more straightforward to order the genes by the mutational components that they are associated with, which will also facilitate reading of the text in the various section in pages 9-11. A side note: please label the colored genes within the figure so the readers can understand the colors without having to read the legend.

Thank you for the suggestions. We wanted to order the genes in a way that functionally similar genes are grouped together to help the reader to see for

instance how *BRCA1/2* and *PALB2* all associated with dHR components. Additionally, the matrix with associations was too sparse to run hierarchical clustering on this visualisation. We agree that this has not been the best approach and reorganised the Figure, so as to show all replicated associations in alphabetic order (Extended Data Figure 5 is now Supplementary Figures 13 & 14). The functional grouping can be interpreted from the network analysis (Figure 4 and Supplementary Figure 19).

3. How to interpret a gene-mutational component association that is significant under multiple models of inheritance (for example, and the intersection area in the Venn diagram in Fig 2f and *BRCA1/2* in Fig 3)? Can the authors use model competition to reduce the possibilities?

As explained in the comments for reviewer 1 & 2 and shown in our additional subsampling analysis (Supplementary Figures 6-11), it would be difficult to make strong statements about the correct model of inheritance of a gene-component pair in our current setup.

4. Association between *SETD2* and MMR deficiency is highlighted in the abstract and discussed on page 10, but this association was detected at a relatively high FDR level (the actual FDR was estimated to be 44%) and not even shown in Fig 3. Please explain why this case is remarkable.

Indeed, this association was only identified at a more permissive FDR, however there is a very strong prior on this association. There is considerable support from previous studies: (1) biochemistry: the *SETD2* protein deposits the histone mark H3K36me3, which can recruit the MSH6 subunit of the MMR complex MutSa to chromatin [PMID: 23622243]; (2) experiments on human cell lines show that removing the *SETD2* protein can cause microsatellite instability [PMID: 23622243] and that it changes the distributions of mutation rates with respect to H3K36me-marked regions [PMID: 29610279]; and (3) In human cancer data tumours with mutations in *SETD2* (double deletion or somatic mutation) have a MMR-associated mutation phenotype as well [28753428]. In our opinion, these prior studies would be very strong evidence to support this MMR association in *SETD2* germline variants as well, and thus we decided to highlight it.

5. The manuscript is a bit long. I hope the authors can consider streamline the text to highlight the most important findings rather than to briefly touch on all “interesting” cases.

To streamline the text, we transferred a few of our observations to the Supplementary Note and a few technical explanations to the methods.

Parts moved to supplements

Inflation Analysis

Genes associating with a somatic feature enriched in brain and liver cancer

Parts moved to methods:

Explanation of FDRs is only in Methods now

6. Page 4: “identifiy” should be “identify”; page 11: “an decrease” should be “a decrease”.

Thank you, this has been corrected.

REVIEWERS' COMMENTS

Reviewer #1 (Remarks to the Author):

The authors have appropriately addressed all my comments.

Reviewer #2 (Remarks to the Author):

In their revisions the authors have adequately addressed my concerns and have substantially improved their manuscript.

Reviewer #3 (Remarks to the Author):

The revision has largely addressed the concerns of all three reviewers. I particularly appreciate the added analysis, including the power simulation and the regression test to establish direction of each association. In my opinion these analyses substantially strengthen the paper.

A few minor questions arise in the revised version, however, and they need to be addressed before publication.

Lines 225-227 & 1418: it just occurred to me that, the “recessive model” does not necessarily detect only recessive cases in a genetics sense. As individuals with rare pLoF variants without somatic LOH mutation are excluded, the model is effectively testing for differences between individuals with two defective copies versus those with none. Barring likely power issue (due to small number of individuals with biallelic inactivation), this test is expected to show a signal as long as the variant has any effect, regardless whether it is dominant, recessive or additive. To support a strictly recessive case, the authors need to demonstrate there is no difference between people with one defective copy vs people with none; the analysis needed is more complicated for partial recessive case... The authors should at least modify how they describe the “recessive model” throughout the manuscript without using the term “recessive”.

In lines 318-319, the authors stated that many repair genes are likely haplosufficient. This can possibly explain why there are many associations identified under the recessive model, but why is this consistent with the higher replication rate of hits under recessive model? Combined with my comment above, I suspect the cases identified by the “recessive model” are not truly genetically recessive, so the haplosufficiency explanation does not even apply.

Fig 3: A few genes are negatively associated with the ploidy and UV_ICA mutational components. Does this mean that presence of deleterious variants in those genes is correlated with decreased ploidy or mutation burden associated with UV_ICA? How to the authors interpret such negative associations and what could be potential mechanistic explanations?

Typos (possibly more):

Lines 278-279: “reductions the sample size”

Line 580: “these genes a set”

Reviewer #1 (Remarks to the Author):

The authors have appropriately addressed all my comments.

Reviewer #2 (Remarks to the Author):

In their revisions the authors have adequately addressed my concerns and have substantially improved their manuscript.

Reviewer #3 (Remarks to the Author):

The revision has largely addressed the concerns of all three reviewers. I particularly appreciate the added analysis, including the power simulation and the regression test to establish direction of each association. In my opinion these analyses substantially strengthen the paper.

We thank the reviewer for noting that the study was strengthened by additional analyses; please see our responses below.

A few minor questions arise in the revised version, however, and they need to be addressed before publication.

Lines 225-227 & 1418: it just occurred to me that, the “recessive model” does not necessarily detect only recessive cases in a genetics sense. As individuals with rare pLoF variants without somatic LOH mutation are excluded, the model is effectively testing for differences between individuals with two defective copies versus those with none. Barring likely power issue (due to small number of individuals with biallelic inactivation), this test is expected to show a signal as long as the variant has any effect, regardless whether it is dominant, recessive or additive. To support a strictly recessive case, the authors need to demonstrate there is no difference between people with one defective copy vs people with none; the analysis needed is more complicated for partial recessive case... The authors should at least modify how they describe the “recessive model” throughout the manuscript without using the term “recessive”.

We agree in principle with the reviewer that our “recessive model”, when significant does not necessarily imply that the genes/variants are strictly recessive-acting. However, the test does enrich for recessive-acting variants/genes (by limiting to cases with somatic LOH). We have now modified the description of this model in the Methods to make this explicit (see Method section ‘Association Testing via SKAT-O’)

In lines 318-319, the authors stated that many repair genes are likely haplosufficient. This can possibly explain why there are many associations identified under the recessive model, but why is this consistent with the higher replication rate of hits under recessive model? Combined with my comment above, I suspect the cases identified by the “recessive model” are not truly genetically recessive, so the haplosufficiency explanation does not even apply.

As the reviewer notes, our recessive model tests whether two defective copies have an effect compared to none (this enriches for recessive-acting genes, but does not absolutely guarantee that the hits will be recessive-acting, as commented above). In contrast, our dominant model tests whether there is already an effect in individuals with at least one affected copy. Thus, if one-copy alteration would be sufficient to generate an effect, it should have in principle been already captured via the dominant model.

While we would abstain from comparing the individual significant results of the recessive *versus* dominant models due to the limited power of this study (see Supplementary Figures 3-11) and as discussed in the previous reviewer response (see response to reviewer 1 point 2), we can certainly see a trend (Supplementary Figure 12e): despite the low number of individuals with biallelic inactivations available for testing in the recessive model, the validation rate is higher with the recessive model by ~2.5-fold in comparison to the dominant model.

The validation rate would be similar between the dominant and recessive model if none of the replicated associations from the recessive model would be truly recessive (assuming similar power). However the replication rate is higher in the recessive model. Thus, we infer that many of the replicated associations are in fact likely to be truly genetically recessive-acting. In other words, the usage of this model to discover truly recessive associations would be justified, bearing in mind that it is not absolutely guaranteed to yield only recessive associations.

Fig 3: A few genes are negatively associated with the ploidy and UV_ICA mutational components. Does this mean that presence of deleterious variants in those genes is correlated with decreased ploidy or mutation burden associated with UV_ICA? How to the authors interpret such negative associations and what could be potential mechanistic explanations?

Indeed, this is the correct interpretation of the results. The replicated associations we provide can provide a starting point for future follow-up work to elucidate mechanisms; some instances of this were further supported by literature review (e.g. *PAXIP1*, *RIF1*, *WRN*) and/or our network analysis (e.g. *MSH3*, *EXO1*, *MLH3* in Figure 4 and Supplementary Figure 20). Potential explanations for negative associations could be, among others, that (1) the loss-of-function in a gene can decrease relative mutation rates e.g. as may be the case for TLS (error-prone) DNA polymerase enzymes or (2) that the germline variant is associated with a tumor subtype which has a low ploidy leading to this correlation/association or (3) that our association test picks up the signal for possible gain-of-function variants in DNA repair genes (of note our analysis currently does not distinguish between those, and other possible mechanistic scenarios). Overall, we conclude that the associations we report here are useful for hypothesis generation for downstream experimental work to validate their causal role and ascertain mechanisms.

Typos (possibly more):

Lines 278-279: "reductions the sample size"

Thank you, typo corrected.

Line 580: "these genes a set"
Thank you, typo corrected.